# The effective connectome over a century of human life
Guoshi Li [1,2], Khoi Minh Huynh [1,2], Kim-Han Thung[1,2], Hoyt Patrick Taylor IV [1,2], Guoye Lin[1,2], Weili Lin [1,2], Sahar Ahmad [1,2] & Pew-Thian Yap [1,2] ✉

Brain functional connectomes describe the coordination principles of neural systems. Understanding normative developmental connectomes is crucial for standardized growth assessment and early disease detection. However, functional connectome has been mostly studied using undirected functional connectivity estimated based on statistical correlation of functional MRI (fMRI) signals. Effective connectivity (EC), in comparison, builds on a generative model of neural interactions and provides directed connectivity strengths among neural populations. To understand how EC evolves with age, we charted the lifespan effective connectome of human brain networks based on high-quality fMRI data from the Lifespan Human Connectome Projects. We found that global and network EC follows an overall inverted U-shape development with an average maturation time at around 9 years of age, significantly earlier than functional connectivity. Regional EC strength exhibits diverse evolution patterns and is more variable during early development than later life, underscoring a critical early window of plasticity. Also, maturation of excitatory and inhibitory nodal EC follows opposite hierarchical sequences. Moreover, the development of nodal EC is constrained by the sensorimotor-association (S-A) gradient, primarily governed by inhibitory EC. Our work reveals fundamental development principles of directed causal interactions between functional networks, offering a foundation for more precise and individualized brain assessments.

Human cognitive functions depend critically on the complex interactions of distributive neuronal systems and circuits, which continuously evolve over the human lifespan[1–6]. Understanding the evolutionary principles of such dynamic interactions is important to decipher the neural basis of cognitive development and decline. The evolution of macroscopic neural interactions can be studied by lifespan connectomics: an emerging field dedicated to unveiling the wiring diagram and coordination principles of neuronal networks over the course of brain development and aging[7]. By integrating network science into the characterization of structural and functional organization of the human brain across different life periods, lifespan connectomics has enhanced our understanding of neurodevelopment, adulthood plasticity, and neurocognitive decline[7–12]. Lifespan connectomics is not only crucial for understanding normal cognitive development and decline but also important for unraveling the pathological basis of neurodevelopmental and neurodegenerative disorders, which are linked to disrupted connectivity patterns at specific brain development stages[13–18]. Understanding when the brain is most vulnerable to pathological disruptions helps pinpoint optimal windows for early intervention.

One powerful toolset for lifespan brain studies is reference growth charts that characterize the developmental trajectories of normative brain properties. Similar to growth charts for physical traits such as height and weight[19], growth charts of the brain are invaluable for standardized assessment of individual growth and early detection of diseases. Two notable studies have recently made great progress in delineating the normative growth trajectories of brain morphology and connectomes: (i) Bethlehem et al.[20] characterized the nonlinear trajectories of brain structural changes by assembling the largest structural MRI dataset to date (123,984 scans from 101,457 participants) across more than 100 primary studies, allowing robust quantification of individual variations in morphology. (ii) Sun et al.[21] charted the normative growth trajectory of functional connectome over the lifespan by aggregating a large multimodal neuroimaging dataset (34,731 resting-state functional MRI (rs-fMRI) and structural MRI scans from 33,250 subjects aged between 32 postmenstrual weeks and 80 years). The study revealed that functional connectivity (FC) peaks in the fourth decade of life and identified a spatiotemporal gradient axis that regulates the development of regional connectivity, greatly advancing our understanding of the growth principles of functional networks. Prior to this comprehensive

[1]Department of Radiology, University of North Carolina at Chapel Hill, Chapel Hill, NC, USA. [2]Biomedical Research Imaging Center, University of North Carolina at Chapel Hill, Chapel Hill, NC, USA. ✉e-mail: ptyap@med.unc.edu

exploration, multiple lifespan studies have been conducted to characterize age-dependent changes in FC, though with much smaller sample sizes (<1000) and inconsistent results[22–26].

Despite the great progress in charting functional connectome over the lifespan, functional connectivity, defined as the statistical dependency among fMRI signal variations, has significant limitations due to its descriptive and undirected nature that cannot provide causal relationships and mechanistic insights into neural system interactions[27]. In contrast, effective connectivity (EC) builds on a generative model of neural interactions and corresponds to the coupling strengths of the neuronal model estimated from the observed neuroimaging data[28,29]. As the generative model describes how the latent (hidden) neuronal states and their interactions give rise to the observed neuroimaging measurements, EC can potentially provide mechanistic neuronal accounts for both normal cognitive processes and abnormal disease states. Specifically, EC models incorporate both excitatory and inhibitory neural interactions, thus able to infer excitation–inhibition (E–I) balance and differentiate the distinct roles of excitatory and inhibitory processing in cognitive development and aging. Thus, EC offers an important metric to quantify neural system interactions complementary to FC, and charting EC over the human lifespan is necessary for a more complete understanding of functional network development and degeneration. Despite the widespread use of EC in human connectomics[30–36], no reference charts have been established so far to characterize EC over the life cycle, although such growth charts are highly desired for benchmarking individual trajectories of functional development.

To close this gap, we delineated EC development of human brain networks across the lifespan based on large-cohort and high-quality rs-fMRI datasets from the Lifespan Human Connectome Projects (HCPs)[37–40]. We assembled 2904 rs-fMRI scans from 2696 healthy subjects from 10 days to 100 years of age (Supplementary Fig. S1). After preprocessing of rs-fMRI data, regional averaged blood oxygenation level-dependent (BOLD) time series were extracted using the Schaefer-7 networks atlas[41] with 400 cortical regions grouped into seven functional networks: visual (VIS), somatomotor (SM), dorsal attention (DAN), ventral attention/salience (VAN/SAL), limbic (LIM), frontoparietal control (FPC), and default mode (DMN) networks[42]. We applied regression dynamic causal modeling (rDCM), a novel variant of DCM for whole-brain EC estimation[43–46], to infer EC among the 400 regions for all subjects. We then utilized a generalized additive mixed model (GAMM) to characterize the nonlinear trajectories of EC at the whole-brain, network, and regional levels.

## Results
### Global and network effective connectivity across the human lifespan
To characterize EC changes over the life cycle, we divided the whole lifespan into seven developmental stages[20]: infancy (0–1 year), early childhood (1–6 years), late childhood (6–12 years), adolescence (12–20 years), young adulthood (20–40 years), middle adulthood (40–60 years), and late adulthood (60–100 years). The average EC matrices during different developmental stages are shown in Supplementary Fig. S2. We observed that brain EC was dominated by excitatory (positive) interactions and exhibited a clear modular structure (i.e., higher within-network than between-network EC) across all seven stages; such a modular structure became more evident starting late childhood. The lifespan curve of the mean global EC across the entire brain displayed an overall inverted U-shape nonlinear trajectory which increased from around 2 years old through late childhood, peaked at around 9 years old (males: 8.7 years, 95% bootstrap confidence interval (CI) 5.6–13.8; females: 8.7 years, 95% bootstrap CI 6.5–10.5), and declined gradually toward late adulthood (Fig. 1a). The growth rate reached its maximum at 4.7 years for males and 4.5 years for females, while the maximal rate of decline (after the peak) occurred at 11.9 years for males and 12.5 years for females. Interestingly, the overall increase-decrease growth pattern was initiated by a dip from birth to around 2 years old, and the females declined and increased faster than males. This indicates substantial differences between males and females in the development of the effective connectome.

Notably, the peak for EC occurred much earlier than that reported for FC (40 years)[21], suggesting a fundamental difference between FC and EC maturation. The growth patterns of the mean global excitatory (positive) and inhibitory (negative) EC are shown in Supplementary Fig. S3. While the profile of the excitatory EC mainly followed the inverted U-shape of the overall EC (compare Supplementary Fig. S3a with Fig. 1a), the inhibitory EC showed an overall U-shape development pattern (Supplementary Fig. S3b). Specifically, the inhibitory EC increased (in absolute value) from birth to 16 years old for males and 49.2 years old for females, followed by a decrease toward later life, indicating more sustained global inhibition in females. Thus, both excitatory and inhibitory global EC obey the same basic development principle that EC first potentiates to a global peak and then declines with age. However, inhibitory EC matures at a later age than excitatory EC.

To understand how EC evolves at the network level, we charted both within-network EC and between-network EC for each functional work. The mean within-network EC and mean between-network EC (among the seven networks) across the lifespan are shown in Fig. 1b and c, respectively. Although both within-network and between-network EC followed the overall inverted U-shape pattern, there were substantial differences: (i) The mean within-network EC was much larger than the mean between-network EC, and the age-dependent increase and decline were much more prominent in the mean within-network EC. This suggests that age has a stronger effect on within-network EC than on between-network EC. (ii) The peak of the mean within-network EC (males: 9.5 years (95% bootstrap CI 7.7–11.6); females: 9.7 years (95% bootstrap CI 7.9–11.7)) occurred later than the mean between-network EC (7.8 years for males (95% bootstrap CI 3.3–11.3) and 8.1 years for females (95% bootstrap CI 5.7–10.2)). (iii) The mean within-network EC was characterized by a larger dip in early age (0–2 years), while such an early dip was much smaller for the mean between-network EC, especially for males. (iv) There was a larger sex difference in the mean between-network EC compared with the mean within-network EC, suggesting that between-network EC, but not within-network EC, contributed mostly to the global developmental sex difference. In particular, the females increased faster but also declined faster than males (Fig. 1b, c, bottom panels).

The evolution of the within-network EC and between-network EC for each individual system is shown in Supplementary Fig. S4. We observed network-specific developmental patterns for both within-network and between-network EC. Specifically, the primary sensory (VIS and SM) and attentional (DAN and VAN) networks showed higher within-network EC than the association systems (LIM, FPC, and DMN) (Supplementary Fig. S4a), while the attentional networks displayed the highest between-network EC than other networks for most of the life cycle (Supplementary Fig. S4c). In addition, the primary sensory and attentional networks showed more prominent early dip (0–2 years) followed by stronger increase than the association systems, which exhibited a mostly decreasing profile throughout lifespan (Supplementary Fig. S4a, c). The larger sex difference in between-network EC than within-network EC was also evident at the individual network level (Supplementary Fig. S4b, d).

As the global EC showed contrasting developmental patterns for excitatory and inhibitory connectivity, we separated the excitatory and inhibitory components for each individual system, and for both within-network EC (Fig. 1d, e) and between-network EC (Fig. 1f, g). Similar to the global EC, the excitatory within-/between-network EC followed an overall inverted U-shape development, while the inhibitory within-/between-network EC exhibited a U-shape evolution pattern for the mean EC and most of the functional networks. A few exceptions included the dorsal attention network that showed a continuous decrease in excitatory between-network EC (Fig. 1f), the ventral attention network that showed an inverted U-shape in inhibitory within-network EC (Fig. 1e), and the limbic network that showed a linear increase in both inhibitory within-network EC and inhibitory between-network EC (Fig. 1e, g). Importantly, we observed more distinct network evolution patterns in excitatory and inhibitory network EC (Fig. 1d–g) compared with the overall network EC (Supplementary Fig. S4). First, the primary sensory networks (VIS and SM) displayed the highest

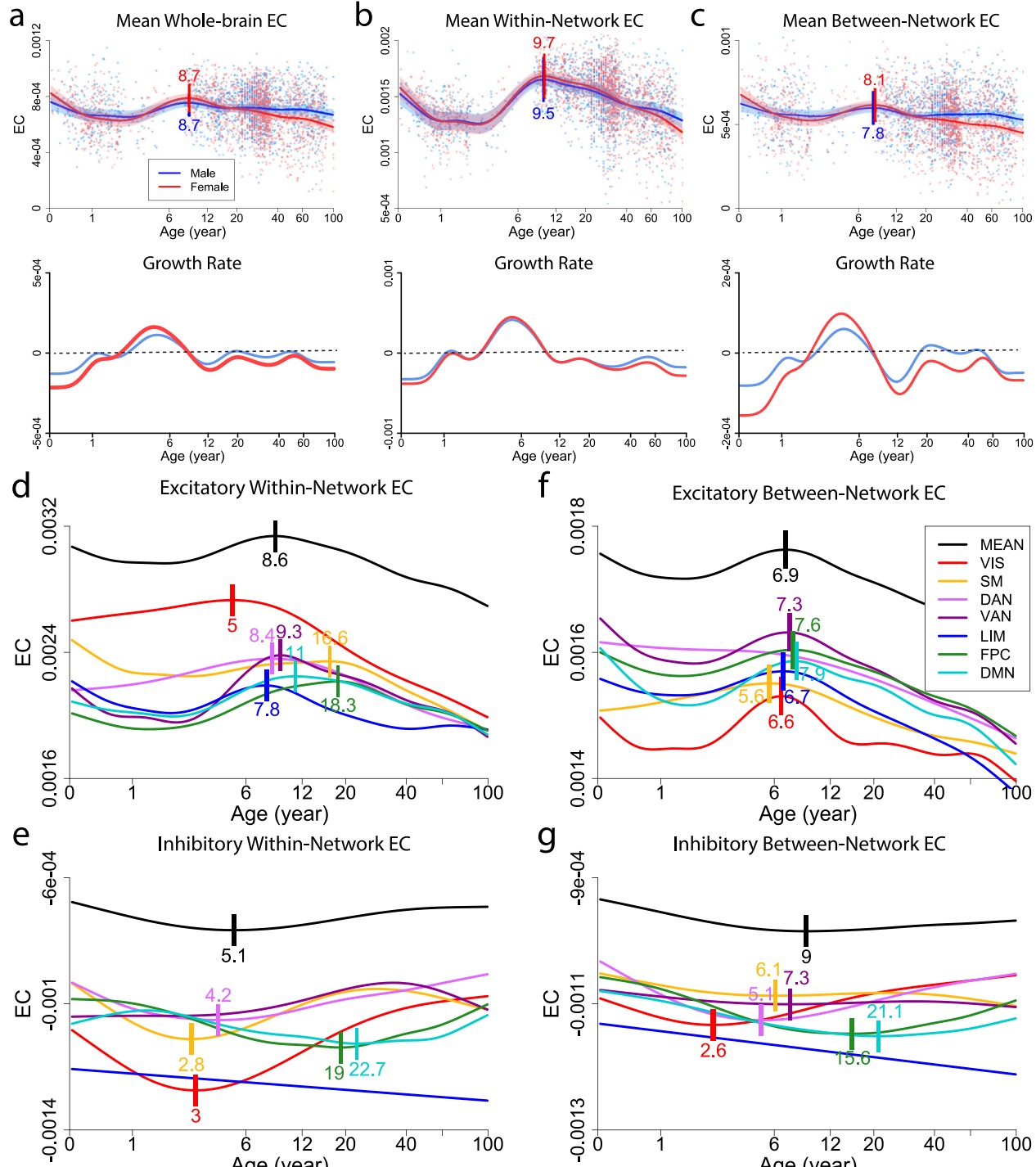

**Fig. 1 | Evolution of global and network effective connectivity (EC) over the lifespan.** Normative trajectories (*top*) and velocity curves (*bottom*) of **a** mean whole-brain EC, **b** mean within-network EC, and **c** mean between-network EC stratified by sex. Background points denote EC metrics of individual subjects as a function of age; points are colored by sex. **d** Developmental trajectories of the mean and network-specific excitatory within-network EC. **e** Developmental trajectories of the mean and network-specific inhibitory within-network EC. **f** Developmental trajectories of the mean and network-specific excitatory between-network EC. **g**, Developmental trajectories of the mean and network-specific inhibitory between-network EC. For (**a**–**g**), the horizontal axis is in log scale, and the short vertical bars indicate the peak ages. For (**a**–**c**), the shaded area denotes the 95% confidence interval of the normative trajectories. For (**d**–**f**), the mean trajectory is manually lifted upward for better visualization. VIS visual network, SM somatomotor network, DAN dorsal attention network, VAN ventral attention network, LIM limbic network, FPC frontoparietal control network, DMN default mode network.

excitatory within-network EC, followed by the two attentional networks (DAN and VAN), and the three association networks (LIM, FPC, and DMN) (Fig. 1d). Such a network ranking switched for the excitatory between-network EC where the two attentional networks exhibited the

highest EC followed by the three association networks, and then the two primary sensory networks (Fig. 1f). Second, the visual network showed the largest peak in inhibitory within-network EC while the control and default mode networks exhibited the largest peaks in inhibitory between-network EC

EC (Fig. 1e, g). Notably, the limbic network exhibited the highest and continuously increasing inhibitory EC for both within-network and between-network connections. Lastly, the primary sensory and attentional networks displayed much earlier peaks (2.6–7.3 years) in inhibitory within-/between-network EC than the two high-order networks (FPC and DMN: 15.6–22.7 years) (Fig. 1e, f), which contributed to a larger earlier dip in the overall within-/between-network EC (Supplementary Fig. S4). Such early maturation in sensory and attentional systems largely held true for the excitatory within-/between-network EC (Fig. 1d, f), though the developmental peaks of all networks were located closer to each other than the inhibitory network EC. Of note, the visual or somatomotor network exhibited the earliest peak for the excitatory and inhibitory network EC. As the two attentional networks can be interpreted as sensorimotor processing systems situated between the primary sensory systems and upstream association networks[47], the development of network EC is organized by the sensorimotor to association (S-A) axis that is characterized by earlier maturation of the sensory/sensorimotor processing systems.

## Lifespan changes in functional segregation and integration

Functional segregation and integration are two basic organizational principles of human brain development[48]. To understand how functional segregation and integration mature with age in the context of EC, we defined both a network segregation index (NSI) and a network integration index (NII) based on within-network EC and between-network EC. The NSI is defined as the difference between within-network EC and between-network EC normalized by within-network EC for a particular network[21], while the NII of a specific network is calculated as the absolute sum of excitatory between-network EC and inhibitory between-network EC (see "Methods" section). Thus, NSI depends on the relative strength between within-network EC and between-network EC, while NII embodies the strength of both excitatory and inhibitory network interactions. The mean NSI increased steadily after birth, peaking at 15.3 years for males (95% bootstrap CI 13.3–17.2) and 20 years for females (95% bootstrap CI 17.2–23.1), followed by a slow decline through adulthood (Fig. 2a). Of note, the peak time of the mean NSI based on EC was earlier than the peak time reported based on FC (23.5 years)[21]. The maximal rate of NSI increase occurred at 6.1 years old for males and 7.6 years old for females. This suggests that functional network segregation measured by EC develops mostly during childhood and stabilizes during adulthood. Interestingly, males showed a faster decline than females after the peak, reflecting more persistent between-network EC in males than females during adulthood (Fig. 1c). By comparison, the mean NII exhibited a more moderate and much shorter increase phase than NSI after birth, which peaked at 6.5 years old for males (95% bootstrap CI 2.6–11.9) and 6 years old for females (95% bootstrap CI 2.3–10.6) before declining gradually thereafter (Fig. 2b). The maximal increase rate took place at 2.8 years old for both males and females. The earlier maturation time of network integration was consistent with the earlier peak age of between-network EC compared with within-network EC (Fig. 1b, c). Interestingly, there was less sex difference in network integration compared with network segregation (compare Fig. 2b with a), as the opposing sex effects on excitatory EC and inhibitory EC could cancel out each other (Supplementary Fig. S3).

The network-specific growth curves of NSI and NII across the entire lifespan are shown in Fig. 2c and d, respectively. We observed network hierarchy in both NSI and NII, which switched between system segregation and integration. Specifically, the two primary sensory networks (VIS and SM) started with the highest NSI and maintained the high level of NSI for most of the life cycle (despite a considerable drop for the SM during early childhood) (Fig. 2c). The two attentional networks (DAN and VAN) started with lower NSI and increased continuously up to adolescence while maintaining their positions in the middle of the hierarchy. The two high-order networks (FPC and DMN) started with the lowest NSI and increased substantially during childhood and adolescence to a level either as high as the primary networks or under the attentional networks. Starting with an intermediate level of NSI, the limbic network first decreased and then

increased to a local peak at 12.6 years, followed by a slight decrease to the bottom of the hierarchy and a subsequent increase during middle and late adulthood (>40 years). The late-life increase is due to the fact that the NSI reflects the difference between within-network EC and between-network EC, and the between-network EC of the limbic network declined while the within-network EC remained stable during middle and late adulthood (Supplementary Fig. S4a, c). On the other hand, the attentional and association networks manifested the highest and intermediate NII levels respectively during early life (0–5 years); such a ranking switched after early childhood (>6 years) with the association networks climbing up to the top of the hierarchy and the attentional networks being in the middle (Fig. 2d). In contrast, the primary sensory networks displayed the lowest NII throughout the lifespan. It is noted that the visual network exhibited the earliest maturation peak for both network segregation and integration, and the sensory and attentional networks matured earlier (4.9–5.5 years) than the association systems (7.7–8.4 years) in network integration (Fig. 2d). By comparison, except the earliest maturation of the visual system (1.5 years), the segregation maturation peaks of other networks occurred during adolescence and young adulthood (12.6–24.3 years) and did not exhibit network dependency (Fig. 2c). In addition, network segregation was more robust to late-age decline (>20 year) than network integration (comapre Fig. 2c with d).

Cortical visualization of NSI and NII at four selected ages (years 1, 6, 21, and 60) is shown in Fig. 2e and f, respectively. The visual and somatomotor networks exhibited a much higher segregation level than other networks in years 1 and 6. As the NSI of other networks increased with age, network segregation became more homogeneous among different networks at years 21 and 60. By comparison, the dorsal and ventral attention networks showed a higher integration level than other networks in year 1, but the contrast was less distinct compared to network segregation at the same age (i.e., year 1). At year 6, the NII of the limbic, control, and default mode networks increased to the same level as the two attentional networks, further reducing the network heterogeneity. At age 21, with the decrease of NII in the sensory and attentional networks, the association systems showed the highest level of NII, and network heterogeneity increased. At age 60, the NII of all networks reduced, but network heterogeneity remained relatively high. Overall, functional segregation and integration represent two opposing and dynamic processes where system segregation becomes more homogeneous, while system integration becomes more heterogeneous among networks with age.

## Heterogeneous developmental patterns of nodal effective connectivity strength

We next examined nodal EC strength (ECS), defined as the sum of all incoming EC to a particular region (corresponding to the sum of each row in the EC matrix). We fitted 400 nodal ECS using GAMM representing the evolution of regional net excitation (i.e., excitation–inhibition) across the life cycle. To characterize the developmental pattern of nodal ECS, we applied K-means to group the 400 nodal ECS time courses into five distinct clusters (Fig. 3a): (i) After an initial dip in the first year, the nodal ECS quickly climbs up to a plateau at around 10 years old and continues to increase slowly up to middle adulthood followed by a late-life decline; (ii) The nodal ECS decreases from birth to around 2 years and shows a slow yet continuous increase up to middle adulthood followed by a late-life decline; (iii) After an early dip in the first one to two years, the nodal ECS reaches a peak at around 9 years old followed by a gradual decline; (iv) With a small early dip during the first one to two years, the nodal ECS maintains relatively stable from birth up to a peak at around 9 years old followed by a relatively fast decline toward late adulthood; and (v) The nodal ECS is characterized by a quasi-linear decline throughout the lifespan.

The distribution of different clusters among the 400 regions is shown in Fig. 3b. We observed that clusters 1 and 5 were the two major patterns (cluster 1: 27.8%, cluster 5: 26%), representing the two opposite ends of the evolution tendency (one continuous increase for most of the life cycle and one continuous decline). The other three clusters showed similar occurrence (cluster 2: 13.3%, cluster 3: 16.3%; cluster 4: 16.8%). At the individual

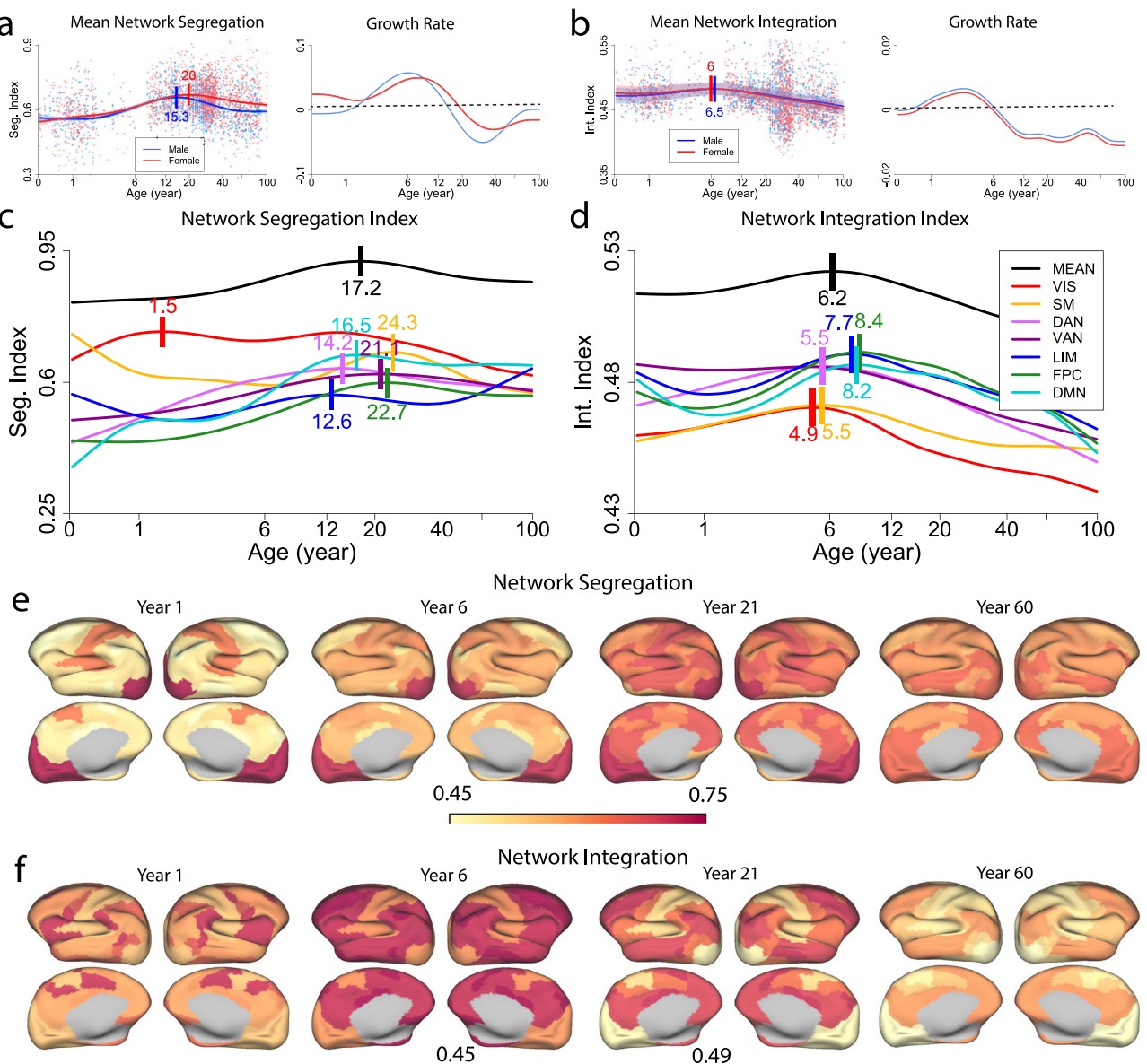

**Fig. 2 | Evolution of network segregation and integration over the lifespan.** Normative trajectories (*left*) and velocity curves (*right*) of **a** mean network segregation index (NSI), and **b** mean network integration index (NII) stratified by sex. Background points denote NSI or NII of individual subjects as a function of age; points are colored by sex. **c** Developmental trajectories of the mean and network-specific segregation index. **d** Developmental trajectories of the mean and network-specific integration index. **e** Cortical visualization of NSI at four selected time points. **f** Cortical visualization of NII at four selected time points. For (**a**–**d**), the horizontal axis is in log scale, and the short vertical bars indicate the peak ages. For (**a**) and (**b**), the shaded area denotes the 95% confidence interval of the normative trajectories. For (**c**) and (**d**), the mean trajectory is shifted upward for better visualization. VIS visual network, SM somatomotor network, DAN dorsal attention network, VAN ventral attention network, LIM limbic network, FPC frontoparietal control network, DMN default mode network.

network level, similar pattern distribution was shared among the sensory and attentional networks (VIS, SM, DAN, and VAN), and among the association systems (LIM, FPC, and DMN) (Fig. 3c). Specifically, cluster 1 was the predominant pattern for the visual network (70.5%), the major pattern for the somatomotor (26%) and ventral attention (36.2%) networks, and the second major pattern for the dorsal attention network (23.9%). Also, cluster 2 was the predominant pattern for the dorsal attention network (50%), and the third major pattern for the somatomotor (20.8%) and ventral attention (17%) networks. On the other hand, clusters 4 and 5 were the two primary patterns for the limbic (cluster 4: 42.3%, cluster 5: 50%) (note that LIM contained only three clusters), control (cluster 4: 23.1%, cluster 5: 36.5%) and default mode (cluster 4: 23.1%, cluster 5: 39.6%) networks. The occurrence of cluster 3 was a bit higher in the association systems (16%) than in the sensory/attentional networks (13.5%). Consequently, the nodal ECS

time courses exhibited the highest correlation within the visual network and the dorsal attention network (each with a predominant cluster), and among the limbic, control and default model networks (with two shared primary clusters) (Fig. 3d). A high level of correlation was also evident among the sensory/attentional systems because of similar cluster distribution. Moreover, the somatomotor and attentional networks (SM, DAN, and VAN) exhibited a moderate level of correlation with the three association systems, mainly due to the common pattern shared by cluster 3, and other clusters to a lesser degree.

The peak age distribution of nodal ECS for each network is displayed in Fig. 3e. Statistical comparison showed that the median peak ages of the visual, somatomotor, dorsal attention and ventral attention networks were significantly higher than each of the association systems (LIM, FPC and DMN) ($Z > 4.1$, Cohen's $d > 0.4$, $p < 0.00005$, FDR corrected, Wilcoxon

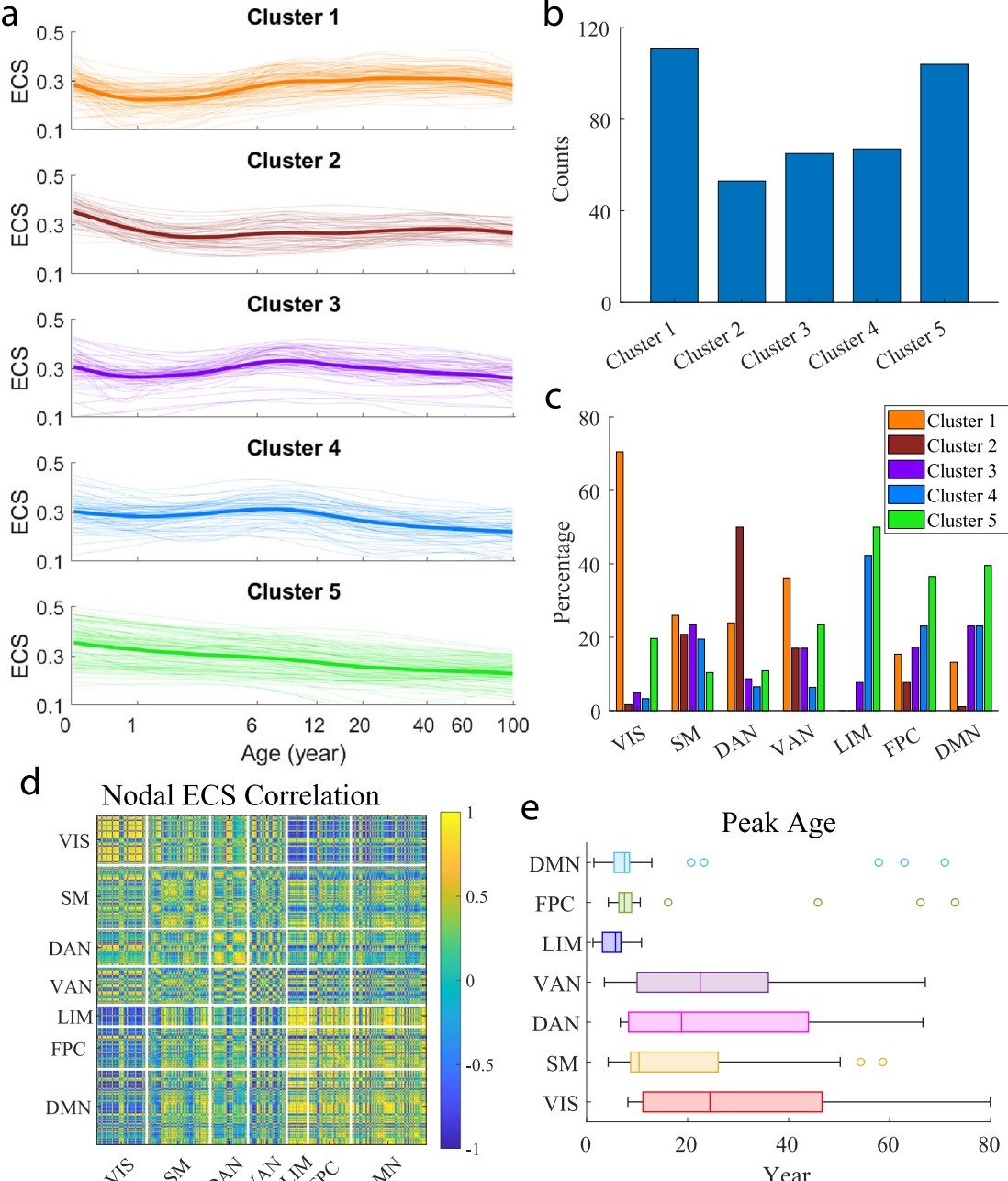

**Fig. 3 | Heterogeneous developmental patterns of nodal effective connectivity strength (ECS). a** Classification of 400 fitted nodal ECS time courses into five different clusters representing distinct temporal patterns. The average nodal ECS profile for each cluster is in bold. **b** Distribution of the five clusters among 400 regions. **c** Percentage of each cluster in each functional network. **d**, Correlation matrix of the 400 nodal ECS time series. **e** Distribution of the peak ages of nodal ECS in each functional network ($n$ = 61, 77, 46, 47, 26, 52, and 91 regions for VIS, SM, DAN, VAN, LIM, FPC, and DMN, respectively). For the boxplots, the central mark indicates the median, and the left and right edges of the box indicate the 25th and 75th percentiles, respectively. The whiskers extend to the most extreme data points not considered outliers, and the outliers are plotted individually using the "o" marker symbol. For **a**, the horizontal axis is in log scale. VIS visual network, SM somato-motor network, DAN dorsal attention network, VAN ventral attention network, LIM limbic network, FPC frontoparietal control network, DMN default mode network.

rank sum test). This is because the sensory/attentional processing networks contained mostly clusters 1 and 2, whose maximal peak ages were located in adolescence or adulthood. By comparison, the three association systems consisted mostly of clusters 3, 4, and 5 whose peak ages resided within childhood. It should be noted that most ECS time courses in cluster 5 had no local peaks due to continuous decrease thus were not included in peak age calculation. Overall, the results suggest that the net excitation of most regions in the sensory/attentional processing networks initially reduces in early life (0–2 years), but then gradually increases and matures in adolescence and adulthood. In contrast, the net excitation of most regions in the association networks either decreases throughout the lifespan or reduces during early life (0–2 years) and then culminates in childhood, followed by a continuous decline in later age.

To delineate the contributing roles of excitatory and inhibitory EC to nodal ECS development, we applied the same clustering analysis to excitatory nodal ECS (Supplementary Fig. S5) and inhibitory nodal ECS (Supplementary Fig. S6), respectively. We observed the same set of clusters for excitatory nodal ECS as the overall nodal ECS, but the existence of

clusters 1 and 2, the two increase-dominant patterns, reduced both globally and in the sensory/attentional networks (compare Supplementary Fig. S5a–c with Fig. 3a–c), suggesting that the inhibitory EC contributed to the increasing patterns. In the meantime, the occurrence of clusters 3–5, the three decrease-dominating patterns, increased both globally and in the sensory/attentional networks. As a result, the excitatory nodal ECS time courses became more homogeneous, as reflected by heightened correlation within and between networks, and the difference in peak ages between the sensory/attentional networks and the association systems reduced substantially compared with the overall ECS (compare Supplementary Fig. S5d, e with Fig. 3d, e).

In contrast, the inhibitory nodal ECS was characterized by a different set of evolution patterns that represented the variations of the U-shape development curve (compare Supplementary Fig. S6a with Fig. 3a). The inhibition decrease-dominating pattern (cluster 1) and the inhibition increase-dominating pattern (cluster 5) denoted the two major patterns in the whole brain (Supplementary Fig. S6b). Although similar pattern distribution was shared among the sensory/attentional networks and the association systems, respectively, the level of similarity was reduced compared with the overall nodal ECS (compare Supplementary Fig. S6c with Fig. 3c). Consequently, the nodal ECS time courses became more heterogeneous, as indicated by a reduced level of correlation within and between networks (compare Supplementary Fig. S6d with Fig. 3d). Interestingly, opposite to the overall or excitatory nodal ECS, the sensory and attentional networks were dominated by the patterns with small (negative) peak ages (i.e., clusters 1–3), while the association networks were dominated by the patterns with large (negative) peak ages (i.e., clusters 4 and 5) (Supplementary Fig. S6c), leading to much lower peak ages in the sensory/attentional networks compared with the association networks (compare Supplementary Fig. S6e with Fig. 3e).

The smaller peak ages of the inhibitory nodal ECS in the sensory/attentional networks compared to the association systems (Supplementary Fig. S6e) were consistent with their earlier maturation peaks in both within-network and between-network inhibitory EC (Fig. 1e, g). However, the largest peak age of the excitatory nodal ECS in the visual network (Supplementary Fig. S5e) seemed to be contradictory to its earliest (or second-earliest) peaks in within-network and between-network excitatory EC (Fig. 1d, f). One major difference was that nodal ECS reflected the sum of only incoming EC to a particular region (Supplementary Fig. S7a), while between-network EC contained both incoming (from other networks to a particular network) and outgoing (from a particular network to other networks) between-network EC (Supplementary Fig. S7b). To examine such potential differences, we charted both incoming and outgoing excitatory between-network EC (Supplementary Fig. S7c, d). Indeed, while the visual network maintained its earliest peak for the outgoing between-network EC, it exhibited the latest peak for the incoming excitatory between-network EC, consistent with its latest maturation for the excitatory nodal ECS. Thus, the developmental maturation of nodal and network EC depends not only on its excitatory or inhibitory nature, but also on its direction, fundamentally different from FC maturation, which has no direction and does not distinguish between excitation and inhibition.

### Evolution of nodal and network effective connectivity strength across the life cycle

To confirm the existence of multiple evolution patterns with different peak ages in each network, we plotted the average nodal ECS profiles within each cluster and for each network (Fig. 4a). As expected, the same cluster of different networks shared qualitatively similar characteristics as defined above. Also, it is evident that the maximal peaks of clusters 1 and 2 could well extend into young (20–40 years) and middle (40–60 years) adulthood. By comparison, the maximal peaks of clusters 3 and 4 are centered around 8–10 years old. Notably, nodal ECS showed more variability during infancy and childhood (<12 years) than adulthood (>20 years). This can be quantified by the mean standard deviation (SD) of the nodal ECS time courses before 12

years old and after 20 years old (Fig. 4b), where the mean SD before 12 years old was significantly higher than that after 20 years old, and for all the networks ($Z > 2.4$, Cohen' $d > 0.75$, $p < 0.05$, FDR corrected, Wilcoxon signed-rank test). This suggests that infancy and childhood are critical brain development stages with the most dynamic shaping of effective network interactions.

The fitted nodal ECS across the entire brain is visualized in Fig. 4c at four representative ages (years 1, 9, 20, and 60). At year 1, it can be observed that nodal ECS in the control and default mode networks was much higher than that in other networks, especially in the dorsal lateral prefrontal cortex (dlPFC), posterior cingulate cortex (PCC), and angular gyrus (AG). At age 9, the nodal ECS in the control and default mode networks potentiated and expanded to other regions, including the medial PFC (mPFC). In addition, the nodal ECS in the visual network and the dorsal part of the somatomotor network greatly increased. The dorsal and ventral attention networks also showed substantial elevation in nodal ECS, which was mostly concentrated in the superior parietal cortex (SPC) and anterior cingulate cortex (ACC). The areas with the lowest nodal ECS included the limbic network and regions located around the lateral sulcus, comprised of the lateral somatomotor network, insular cortex, temporal cortex, and supramarginal gyrus. At age 20, the nodal ECS in the control and default mode networks decreased considerably, with little change for other networks. At age 60, the nodal ECS in the default mode network continued to weaken, and the somatomotor and ventral attention networks also exhibited moderate reduction, while the nodal ECS in the visual, dorsal attention, and control networks remained relatively stable.

To further elucidate the developmental changes of nodal ECS at the network level, we charted the lifespan trajectories of network ECS (i.e., the average nodal ECS within a network fitted by GAMM across the lifespan) in Fig. 4d. The overall (mean) trajectory displayed a large sag before 6 years old, peaked at around 8 years old, and declined thereafter. Although most networks followed this overall trend, there were substantial differences. First, the visual and dorsal attention networks showed the largest early sag (measured by the drop from the initial value to the lowest point), followed by the somatomotor, control, and default mode networks. By comparison, the ventral attention and limbic networks displayed little sag before 6 years old. Second, the ECS of the visual, dorsal attention, and control networks were more persistent than other networks during young and middle adulthood (20–60 years). Third, the ventral attention network showed the least change, while the limbic network exhibited the lowest ECS level throughout the lifespan. Lastly, the network ranking switched after 12 years old. Before 12 years old, the control and default mode networks topped most of the other networks on average, and the visual and somatomotor networks lay close to the bottom of the ranking (i.e., just above LIM). After 12 years old, the visual and dorsal attention networks climbed up to the top of the ranking, followed by the ventral attention and somatomotor networks, and the control and default mode networks switched to the low end of the ranking next to the limbic network. This suggests that network excitation is not static, but dynamically evolves across the life cycle. While the higher-order networks receive more excitation during infancy and childhood, the sensory and attentional processing systems have a higher excitation level during adulthood. The normative trajectories of individual network ECS stratified by sex are shown in Fig. 4e. We observed that the association networks showed earlier peaks (6.2–8.4 years) than the sensory/attentional processing systems (8.3–53.5 years), as explained earlier. Also, the largest sex-dependent difference existed during adulthood (>20 years), where the females declined faster than males, consistent with the mean global and network EC (Fig. 1a–c).

### Gradient of effective connectome

It is well known that FC exhibits a gradient, which extends from the primary sensory systems to the association cortex[49]. To investigate whether EC exhibits a similar gradient, we applied principal component analysis (PCA) to the time courses of the fitted nodal ECS across the lifespan[21]. A clear

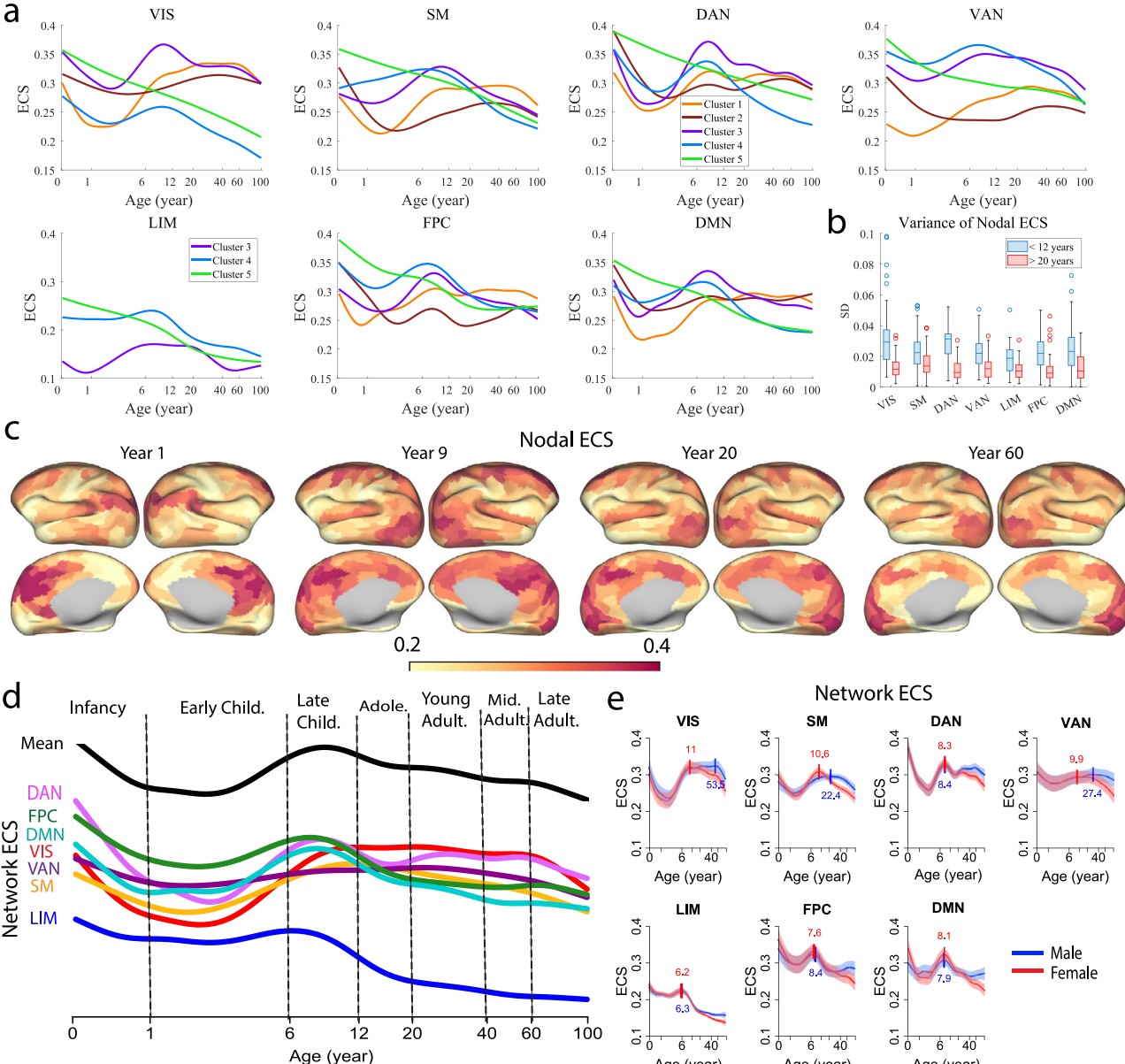

**Fig. 4 | Evolution of nodal and network effective connectivity strength (ECS) over the lifespan. a** Average nodal ECS for each cluster in each functional network. **b** Distribution of the standard deviation (SD) of nodal ECS profiles within the first 12 years and after 20 years old for each functional network ($n = 61, 77, 46, 47, 26, 52$, and 91 regions for VIS, SM, DAN, VAN, LIM, FPC, and DMN, respectively). **c** Nodal ECS visualized in the cortex at four selected time points (years 1, 9, 20, and 60). **d** Developmental trajectories of the mean and network-specific ECS. The mean trajectory is shifted upward for better visualization. **e** Normative trajectories of network ECS for each individual network stratified by sex. The shaded area denotes the 95% confidence interval of the normative trajectories, and the short vertical bars indicate the peak ages. For (**a**), (**d**), and (**e**), the horizontal axis is on a log scale. For the boxplots in (**b**), the central mark indicates the median, and the bottom and top edges of the box indicate the 25th and 75th percentiles, respectively. The whiskers extend to the most extreme data points not considered outliers, and the outliers are plotted individually using the "o" marker symbol. VIS visual network, SM somatomotor network, DAN dorsal attention network, VAN ventral attention network, LIM limbic network, FPC frontoparietal control network, DMN default mode network.

spatial gradient pattern can be observed along the developmental axis of brain EC represented by the first principal component of the PCA (Fig. 5a). Specifically, the visual network exhibited mostly positive loading, the ventral attention network showed slightly more positive loading, the somatomotor and dorsal attention networks displayed a balanced mix of positive and negative loading; whereas, the three association systems (LIM, FPC, and DMN) exhibited mostly negative loading (Fig. 5a, top). Consequently, the developmental axis started from the positive end in the primary visual network, decreased through the ventral attention network, and then the somatomotor and dorsal attention networks, to the high-order networks (FPC and DMN), and reached the negative end in the limbic network

(Fig. 5a, middle). Such a hierarchical spatial transition was evident in the brain surface heat plot (Fig. 5a, bottom).

To understand such a hierarchy at the network level, we plotted the network-averaged nodal ECS trajectories in Fig. 5b. We observed that the network-averaged nodal ECS trajectories were similar to the network ECS profiles (Fig. 4d), with the ventral attention network showing a larger early sag within the first two years. Consequently, the developmental EC gradient was encoded in the network-averaged nodal ECS time courses. Specifically, for the six networks that exhibited early sag, the lowest dig (point) increased from the visual network to the somatomotor network, to the ventral and dorsal attention networks, and then

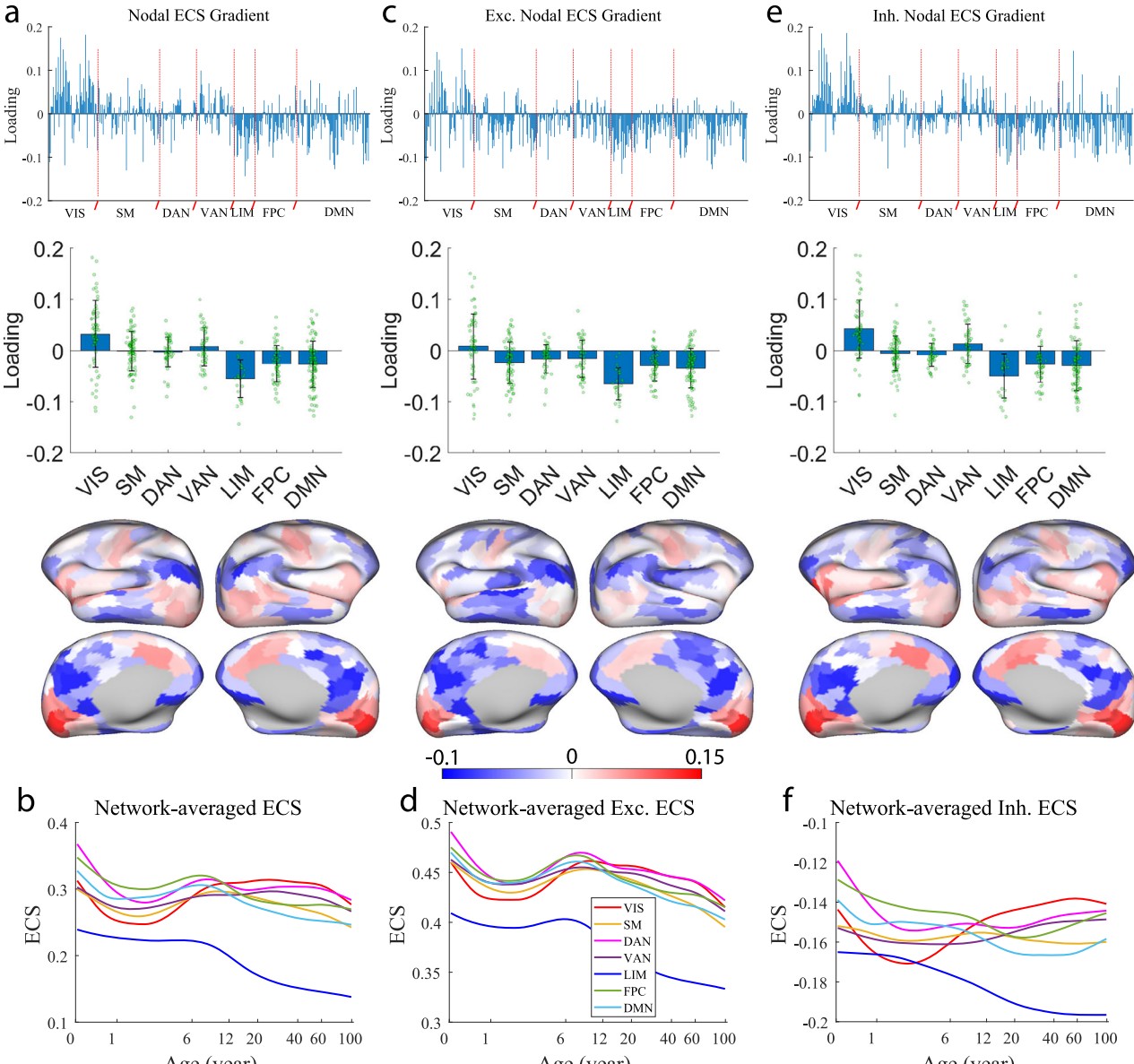

**Fig. 5 | Gradient of effective connectome. a** Nodal ECS gradient. *Top*: First principal component (loading) from the principal component analysis (PCA) on nodal ECS trajectories; *Middle*: Network-averaged loading with individual data points; *Bottom*: Cortical visualization of nodal ECS loading. **b** Network-averaged nodal ECS trajectories. **c** Excitatory nodal ECS gradient. *Top*: First principal component from the PCA on excitatory nodal ECS trajectories; *Middle*: Network-averaged loading with individual data points; *Bottom*: Cortical visualization of excitatory nodal ECS loading. **d** Network-averaged excitatory nodal ECS trajectories. **e** Inhibitory nodal ECS gradient. *Top*: First principal component from the PCA on inhibitory nodal ECS trajectories; *Middle*: Network-averaged loading with individual data points; *Bottom*: Cortical visualization of inhibitory nodal ECS loading. **f** Network-averaged inhibitory nodal ECS trajectories. For (**a**), (**c**), and (**e**) (middle), the error bars denote standard deviation. For (**b**), (**d**), and (**f**), the horizontal axis is plotted on a log scale. VIS visual network, SM somatomotor network, DAN dorsal attention network, VAN ventral attention network, LIM limbic network, FPC frontoparietal control network, DMN default mode network.

to the default mode and control networks (Fig. 5b). By comparison, during adulthood (>20 years), the network-averaged ECS decreased from the visual and dorsal attention networks to the ventral attention network, and then to the somatomotor, control and default mode networks (Fig. 5b). As a result, starting from around 2 years old, the visual network showed the largest increase to adulthood, revealing its positive end in the gradient. The ventral attention network showed a slight increase to adulthood, explaining its small (average) positive loading in the gradient. The somatomotor and dorsal attention networks displayed little change from childhood to adulthood, elucidating their near-zero average loading (Fig. 5a, middle). In contrast, the two high-order networks (FPC and DMN) exhibited a largely decreasing profile from 2 years old, which underlay the large negative loading. Lastly, the limbic

network displayed the most prominent decreasing profile throughout the lifespan, explaining its negative end in the gradient.

When the nodal ECS was calculated from positive (excitatory) EC only, the developmental gradient became less prominent as the loading of the somatomotor and two attentional networks was similar to that of the control and default mode networks (Fig. 5c) and the network-averaged excitatory ECS trajectories became more homogeneous (Fig. 5d). In contrast, if the nodal ECS was computed from negative (inhibitory) EC only, the developmental gradient became even more prominent in that the visual and ventral attention networks showed larger positive loading on average (compare Fig. 5e with a). Consistently, the network-averaged inhibitory ECS trajectories exhibited more distinct evolution patterns (Fig. 5f) where the visual and ventral attention networks showed stronger increasing profiles

(decreased inhibition) after 2 years old, and the control and default mode networks exhibited more prominent decreasing profiles (increased inhibition) from birth up to young/middle adulthood (compare Fig. 5f with b). Thus, the gradient of the EC connectome stems largely from the inhibitory interactions.

## Discussion

By capitalizing on large cohorts and high-quality rs-fMRI datasets from HCP Lifespan studies, we systematically characterized the developmental trajectories of the effective connectome. Our major findings include: (i) Global and network EC follows an overall inverted U-shape development curve with an average maturation time at around 9 years of age. (ii) Primary sensory systems (VIS and SM) show earlier and greater network segregation, while association networks (LIM, FPC, and DMN) exhibit stronger later functional integration. (iii) Nodal ECS exhibits diverse network-dependent evolution patterns, with the sensory/attentional networks dominated by overall increasing patterns and the association networks dominated by overall decreasing patterns. (iv) Nodal ECS demonstrates larger variability during early life (<12 years) than later life (>20 years), highlighting a period of early plasticity. (v) Excitatory and inhibitory nodal ECS follow different developmental patterns with distinct network-dependent maturation peaks. (vi) The development of nodal ECS is constrained by the sensorimotor-association (S-A) gradient, primarily governed by inhibitory EC. These findings add to the existing knowledge of lifespan changes of structural and functional connectome, contributing to a better understanding of brain development and aging.

The overall inverted U-shape development trajectory has been consistently observed in the growth patterns of gray matter (GM), white matter (WM), and functional connectome[21,24,50–53]. Though other global or network-level FC development patterns have been reported, including linear increase or decrease and nonlinear U-shape, such studies typically did not consider the full lifespan cycle or were limited to relatively small sample sizes (<1000)[22,23,26,54]. Also, we observed a relatively early maturation peak around 9 years of age for EC, parallel to the brain's morphological maturation. Indeed, brain weight and intracranial volume experience remarkable increases during the first 6 years of life, followed by continued increase over the next 6 years, albeit at a slower rate, before declining from late adolescence[55,56]. Similar early development of GM and WM volumes has also been observed[20,57,58], and a recent study showed the total GM volume peaked at around 10 years old[59]. In comparison, the peak of FC maturation was reported to occur at a much later age (30–40 years)[21,24], while some studies revealed continuous FC decline across the lifespan[23,60]. Thus, the early maturation of EC likely reflects the early development of brain structure. Notably, an early dip (0–2 years) in EC development is detected at global, network, and regional levels, especially in primary sensory networks. Such an early EC decline represents the early enhancement of inhibitory EC, but also the complex neurobiological processes that take place during infancy and early childhood, including myelination, synaptogenesis, and axonal and synaptic pruning[61] (note that excitatory EC also exhibit an early dip). In particular, the reconfiguration and reorganization due to sensory exposure could result in retraction of callosal fibers up to 70% in primates[62], which may explain the early EC reduction in primary sensory systems. Moreover, on the whole-brain scale, we revealed that females have more prominent early sag and late-age decline than males, which is contributed mostly by differences in between-network EC. This suggests that sex difference in the lifespan connectome arises mainly from long-range neural interactions. Of note, the mean value for the whole-brain EC is relatively low. This is because EC scales with the network size (i.e., larger network leads to lower EC value) and follows a log-normal weight distribution where the predominance of weak connections reflects the low-frequency influences conforming with the fMRI data dominated by low-frequency components[46].

One intriguing finding of our study is that the primary sensory systems and association networks play contrasting roles in functional segregation and integration. Specifically, the primary sensory systems (VIS and SM) showcase greater within-network EC than the association networks (LIM,

FPC, and DMN), while the latter displays higher between-network EC than the former. The network segregation index (NSI), computed as the normalized difference between within-network EC and between-network EC, is characterized by a high level of robust early segregation for the visual and somatomotor networks, followed by a gradual enhancement of the attentional and association networks. Our results are consistent with newborn studies showing that the primary sensory motor systems are largely established at birth and exhibit rather stable connectivity during the first year of life[58,63]. In contrast, higher-order networks remain immature before birth and continue to strengthen after birth to achieve adult-like network architecture[58,63].

On the other hand, the association networks present the highest network integration index (NII) after early childhood. The highest absolute sum of excitatory and inhibitory between-network EC associated with the association networks highlights their importance in cross-network communication and integration, consistent with the functional roles of these association systems[64]. Interestingly, the attentional networks showcase the highest between-network EC, likely reflecting their intermediate position in the network hierarchy to integrate sensorimotor information from lower sensory networks and attentional information from higher-order systems[65,66]. Of note, previous studies tend to converge on the notion that the older brain is less modular and more integrated with decreased within-network FC but increased between-network FC[23,67,68]. By comparison, we observed that functional networks remain well segregated in late adulthood (i.e., persistent NSI) due to the paralleled decline of both within-network and between-network EC, while network integration reduces gradually in the aging brain, reflecting the decline of between-network EC. Again, this indicates that network EC evolution parallels the development of brain morphology and structural connectivity, which show a decreasing profile in the later adult life[20,23,52,59].

At the regional level, we found that nodal effective connectivity strength (ECS) does not follow the same growth trajectory, but displays diverse evolution patterns, consistent with the existence of various developmental shapes of FC strength (FCS) at the regional basis[21,24,69]. Using a clustering algorithm, we were able to cluster the 400 nodal ECS time courses into five distinct patterns. Except for the continuously decreasing pattern and excluding the initial dip, all other four patterns can be viewed as different variations of the inverted U shape with different peak ages and different increasing/decreasing speeds. Despite the various growth curves, nodal ECS trajectories show the highest variability during infancy and childhood while exhibiting a more stereotypical increase or decline during adulthood. This demonstrates that early life, including infancy and childhood, is endowed with a high level of connectome plasticity, forming a critical window for early intervention of neurological and psychiatric disorders.

Notably, we revealed that the distribution of different evolution patterns of nodal ECS in the brain is network-dependent in that the sensory and attentional networks contain mostly the increase-dominating patterns with large peak ages in adolescence and adulthood, while the association systems comprise mostly the decrease-dominating patterns with small peak ages in childhood. As a result, nodal ECS matures much earlier in the association systems than the sensory and attentional networks. We further elucidated that excitatory nodal ECS contains the same five evolution patterns with similar network distribution as the overall nodal ECS. In contrast, the inhibitory nodal ECS consists of a different set of evolution patterns that represent variations of the U-shape development curve. Interestingly, opposite to the overall or excitatory nodal ECS, the sensory and attentional networks include mostly the inhibition decrease-dominating patterns with small peak ages during childhood, while the association networks contain mostly the inhibition increase-dominating patterns with large peak ages in adolescence and young adulthood. Thus, paradoxically, in the sensory and attentional networks, the excitatory nodal ECS matures slowly while the inhibitory nodal ECS matures fast. By comparison, in the association networks, the excitatory nodal ECS develops early while the inhibitory nodal ECS cultivates slowly. Taken all together, nodal ECS displays diverse

evolution patterns that are determined by both the specific network and the excitatory or inhibitory nature of the connections.

Due to the differential maturation of excitatory and inhibitory nodal connections, network excitaiton (or ECS) ranking switches at around 12 years old between sensory/attentional networks and higher-order networks. Specifically, before 12 years old, the control and default mode networks show higher ECS than the sensory/attentional networks. After 12 years old, the sensory/attentoinal networks switch to the top of the ranking with more persistent ECS, while the high-order networks decline to the bottom of the ranking. Such a dynamic switching is consistent with the increase-dominating nodal ECS patterns in the sensory/attentional systems and the decrease-dominating nodal ECS patterns in the higher-order networks. It also reflects slow excitation maturation with fast inhibition development in the sensory/attentional systems, and fast excitation development with slow inhibition maturation in the higher-order networks. Of note, the limbic network showcases the lowest level of ECS with faster decay, consistent with large and continuously increasing inhibitory within-network and between-network EC. As the limbic network is particularly important for emotional processing and memory formation[70,71], weak excitation may be an intrinsic mechanism of the brain to prevent over-reaction to negative stimuli such as fear and anxiety.

Our findings of opposite hierarchical maturation sequences of excitatory versus inhibitory nodal connections are largely consistent with existing experimental observations, but also extend our understanding of the developmental maturation of excitation–inhibition (E–I) balance to the whole lifespan. It is well accepted that human brain development follows a hierarchical sequence along the S-A axis, where sensory systems mature earlier than the association networks[72,73]. Such hierarchical development receives strong and convincing support from animal and human studies. Indeed, maturation of the inhibitory GABAergic circuits occurs fast in the sensory systems, while it takes a protracted course up to adolescence in association networks[74–76], consistent with our model predictions. The sequential development of inhibitory circuitry leads to reduced E–I ratio in the association cortex during adolescence[77–79] and gives rise to the critical plasticity theory, which hypothesizes that heightened inhibition results in increased signal-to-noise ratio, facilitating enhanced experience-dependent plasticity with long-term impact on behavior[72,76,80]. Our results further suggest that maturation of GABAergic inhibition could go beyond adolescence to young adulthood in certain association areas, extending critical plasticity to later life stages.

Compared with inhibitory maturation, it is less clear whether maturation of excitatory connections follows the same sensory to association sequence. Our findings reveal a reverse association to sensory sequence for excitatory connection maturation. The development mismatch between excitation and inhibition may strengthen the E–I balance change, thus promoting the window of critical plasticity. Such a reverse sequence is also supported by experimental and computational evidence including larger increase in basal dendritic tree size and spine density of the excitatory pyramidal cells in prefrontal association cortices compared to primary sensory regions during the first few years of life[81,82], protracted pruning of excitatory synapses during adolescence in association areas relative to sensory regions[73,83] (indicating excitatory decline in association networks during adolescence), and an increase of the E–I ratio in sensorimotor areas during adolescence[78] (suggesting continuous development of excitaiton in sensory systems). In particular, the relatively late maturation of excitatory connections in the primary visual network is consistent with the late progression of visual spatial integration in humans, subserved by long-range spatial interactions[84] as well as the protracted development of glutamatergic proteins in human visual cortex up to 40 years old[85]. It should be noted that nodal ECS reflects the sum of incoming EC to a particular region. On the network level, the visual network exhibits the earliest (or second-earliest) maturation peak for within-network and between-network excitatory EC (Fig. 1d, f). Such discrepancy (between regional and network maturation) can be explained by the fact that incoming between-network EC (consisting of incoming nodal EC) to the visual network displays more

prolonged maturation than outgoing between-network EC (Supplementary Fig. S7c, d), as discussed above.

Lastly, we investigated the hypothesis that the development of an effective connectome over the lifespan conforms to the cortical hierarchy defined by the S-A axis, a robust phenomenon observed during FC development[21,69]. Indeed, the developmental trajectories of nodal ECS form a continuum of growth patterns (or gradient) that extends from the sensory/sensorimotor systems to the transmodal association networks. Specifically, the primary visual network exhibits the strongest increase in average ECS from childhood to adulthood, the somatomotor and two attentional networks display slight increase or little change in average ECS from childhood to adulthood, the control and default mode networks show large ECS decrease from childhood to adulthood, while the limbic network presents the most consistent and prominent decrease in ECS throughout the lifespan. Of note, such network-dependent nodal ECS development is in line with nodal FC strength (FCS) development, where the FCS of the primary sensory regions increases while that of the association networks decreases from childhood to adolescence[21,69]. As the two attentional networks can be interpreted as sensorimotor processing systems[47], the S-A gradient thus traverses from the primary visual network to the sensorimotor processing systems, then to the higher-order networks, and ends with the transmodal limbic network.

Importantly, we found that the developmental gradient depends on the excitatory or inhibitory nature of the effective connectome. When nodal ECS is computed from excitatory EC only, the gradient is reduced as the visual network displays less increase from childhood to adulthood, and the three sensorimotor processing networks (SM, DAN, and VAN) exhibit an overall decreasing profile, as do the association networks. Consistently, the nodal ECS time courses become more homogeneous, reflected by heightened correlation. In contrast, when nodal ECS is computed from inhibitory EC only, the gradient is strengthened as the visual network shows a stronger increasing profile from childhood to adulthood, while the association networks display a more prominent decrease. In agreement with the enhanced gradient, the nodal ECS profiles become more heterogeneous, indicated by a reduced level of correlation. Thus, our results suggest that the gradient of effective connectome stems largely from the inhibitory interactions.

Our findings are consistent with the functional connectivity gradient mapped into a 2D multiscale cortical wiring space where the sensory networks are primarily located in the lower (visual) and upper (somatomotor) left boundaries while the transmodal default mode, frontoparietal, and limbic networks are located more toward the right end with the two attentional networks positioned in the intermediary spot in the wiring space[86]. As the cortical wiring space established based on microstructure proximity and white matter fibers reflects microcircuit features including pyramidal neuron depth and glial expression, such large-scale mapping suggests that the EC development gradient may be determined by the underlying cytoarchitectonic and cellular microcircuit properties. Importantly, directed coherence analysis reveals that inhibitory neuron expression is the most important cellular feature in underpinning the large-scale cortical architecture[86], supporting the critical role of inhibitory connections in shaping the developmental gradient of the effective connectome.

## Conclusions
In summary, we created robust reference growth charts of effective connectome at the whole-brain, network, and regional levels across the entire human life cycle, bridging an important gap in lifespan connectomics. These reference growth charts provide a comprehensive characterization of the development, maturation, and aging processes in terms of directed causal interactions of neural systems, offering a highly valuable toolset to benchmark individual growth trajectories and aid in the early detection of brain disorders.

## Methods
### Imaging datasets
We aggregated 2904 rs-fMRI scans from 2696 subjects (males/females: 1232/1464; age: 10 days to 100 years; Supplementary Fig. S1) from the

Lifespan HCP studies. All the subjects are healthy and cognitively normal, who represent the typical developing and aging samples and are free of major neurological and psychiatric disorders[37–39]. In particular, for subjects 60 years and older, cognitive assessments were conducted to exclude participants with impaired cognitive abilities[39]. We included 263 subjects from Baby Connectome Project (BCP)[40] (471 longitudinal scans; males/females: 124/139; age: 10 days to 6 years, mean: 1.4 ± 1.0 years), 632 subjects from HCP-Development[37] (males/females: 294/338; age: 8.1–21.9 years, mean: 14.7 ± 3.9 years), 1079 subjects from HCP-Young Adult[38] (males/females: 495/584; age: 22–37 years, mean: 28.8 ± 3.7 years), and 722 subjects from HCP-Aging[39] (males/females: 319/403; age: 36–100 years, mean: 60.3 ± 15.7 years). The rs-fMRI data for each subject were acquired from two scan sessions with two runs in each session (anterior to posterior and posterior to anterior); each run lasted for about 7 min for BCP subjects and 15 min for HCP subjects. The time repetition (TR) is 720 ms for BCP and HCP-Young Adult (HCP-YA) scans, and 800 ms for HCP-Development (HCP-D) and HCP-Aging (HCP-A) scans. We included the first run in the first session only for the lifespan EC analysis. The Lifespan HCP studies were approved by the respective local ethics committees, and written informed consent was obtained from the participants (or their guardians). All ethical regulations relevant to human research participants were followed.

## Data processing

To minimize non-biological variations caused by preprocessing discrepancies, we applied a unified preprocessing pipeline to all cohorts. The pipeline begins with a one-step resampling that combines head motion correction with FSL mcflirt, EPI distortion correction with FSL TOPUP using a pair of reverse phase-encoded FieldMaps, rigid (6 DOFs) registration of the SBref image to FieldMaps, and boundary-based registration (BBR) of the distortion-corrected SBref to the corresponding T1w. Subsequent steps include high-pass filtering (cutoff = 0.001 Hz) to remove slow drifts and ICA-AROMA denoising[87] with 150 independent components to eliminate residual motion artifacts. We did not apply low-pass filtering as recent studies have shown that meaningful neural signals can exist in higher frequency ranges (0.1–0.25 Hz)[88]. Additionally, ICA-AROMA denoising uses high-frequency contents as one of the criteria to remove motion-related components, reducing the need for explicit low-pass filtering.

We then used our in-house method established from the work of Ahmad et al.[57] to reconstruct the surface for all scans, resulting in a consistent coordinate system with one-to-one vertex correspondence across the lifespan. The volumetric fMRI data are then mapped to the native cortical surfaces using ribbon-constrained volume-to-surface mapping available in Connectome Workbench. Of note, since our analysis is performed on the cortical surface using vertex-wise measures with subjects' surfaces having one-to-one vertex correspondence, conducting the analysis in native space preserves anatomical precision, avoids misalignment from volumetric warping, and eliminates interpolation artifacts. Thus, our surface-based approach enables accurate group-level comparisons without transforming data to a common voxel grid.

After data processing, we extracted the BOLD time series from the fMRI data using the Schaefer-400 atlas[41], one of the most commonly used parcellation schemes in neuroimaging[89]. We chose the Schaefer atlas as it provides a robust, replicable, and neurobiologically informed framework for analyzing brain functional connectivity and networks[41,90]. Another advantage of the Schaefer atlas is that the 400 cortical regions of interest (ROIs) have been grouped into seven functional networks including the visual ($N = 61$), somatomotor ($N = 77$), dorsal attention ($N = 46$), ventral attention ($N = 47$), limbic ($N = 26$), frontoparietal control ($N = 52$) and default mode ($N = 91$) networks, according to the canonical 7-networks parcellation[42].

## Quality control

We performed quality control (QC) for all cohorts by evaluating the framewise displacement (FD) computed from motion parameters obtained during head motion correction. We excluded samples with mean Power's FD (absolute sum of motion parameters) exceeding 0.5 mm—a common

threshold in fMRI studies[91]. While BCP subjects naturally had higher FD due to baby motion and showed a higher exclusion rate, there was no significant difference among different cohorts after QC (Supplementary Fig. S8). Also, of the QC-passed scans (FD < 0.5 mm), there was a moderate correlation ($R = -0.3$) between FD and age (FD is higher when age is smaller), but the correlation was not significant. Following FD screening, we visually evaluated skull stripping, tissue segmentation, and fMRI-to-T1w image registration. Subjects with problematic or missing cortical surface meshes were also excluded during volume-to-surface mapping of the fMRI data.

## Regression DCM

We use regression dynamic causal modeling (rDCM) for rs-fMRI[46] to estimate the EC among 400 cortical brain regions, generating a 400 × 400 matrix for each fMRI scan. rDCM is a novel variant of DCM that allows efficient estimation of whole-brain effective connectivity[43–46]. Compared to the original DCM[28] or other variants of DCM[92,93], rDCM has the great advantages of high computational efficiency, capable of inverting a whole-brain network model with hundreds of nodes in a few minutes, making it ideal for big dataset applications. Compared to biophysical network models[47,94,95], rDCM does not rely on structural information and estimates individual EC at the single-subject level in addition to its computational superiority. rDCM uses a linear differential equation to describe the neuronal state[43]:

$$\frac{dx}{dt} = Ax + Cu, \tag{1}$$

where $A$ denotes the effective connectivity among different regions and $C$ represents the influence of external inputs $u$ on neuronal activity $x$. rDCM then translates the differential equation to the frequency domain and makes use of the differential property of the Fourier transformation to arrive at the following algebraic equation:

$$i\omega\hat{x} = A\hat{x} + C\hat{u}, \tag{2}$$

where the $\hat{\cdot}$ symbol denotes the Fourier transform, $i$ is the imaginary unit, and $\omega$ is the Fourier coordinate. By replacing the nonlinear hemodynamic model in the original DCM[28] with a linear hemodynamic response function (HRF) and multiplying Eqn. (2) with the Fourier transform of the HRF, we obtain the following equation:

$$i\omega\hat{y}_B = A\hat{y}_B + C\hat{h}\hat{u}, \tag{3}$$

where $\hat{y}_B = \widehat{h * x} = \hat{h}\hat{x}$ is the deterministic (noise-free) prediction of the BOLD signal. To account for the discrete nature of the measured BOLD signal, rDCM applies a discrete Fourier transform (DFT) to convert the continuous BOLD representation into a discrete BOLD representation in the frequency domain, with the following discretization of frequency and time:

$$i\omega := i\mathbf{m}\Delta\omega = 2\pi i \frac{\mathbf{m}}{NT}, \tag{4}$$

where $N$ denotes the number of data points, $\mathbf{m} = [0, 1, …, N-1]$ is the frequency index, and $T$ is the time interval of consecutive data points. Using linear approximation to an exponential function (i.e., difference operator of the DFT), we have:

$$i\omega := 2\pi i \frac{\mathbf{m}}{NT} \approx \frac{1}{T}(e^{2\pi i \frac{\mathbf{m}}{N}} - 1), \tag{5}$$

Substituting Eqn. (5) into Eqn. (3) gives:

$$(e^{2\pi i \frac{\mathbf{m}}{N}} - 1)\frac{\hat{y}_B}{T} = A\hat{y}_B + C\hat{h}\hat{u}, \tag{6}$$

where the hat symbol now indicates the discrete Fourier transform. Next, by assuming the measurement noise to be white noise, the observation model is given by:

$$y_i = y_{B,i} + \epsilon_i, \quad \epsilon_i \sim \mathcal{N}(0, \sigma_i^2 I_{N \times N}) \qquad (7)$$

where $y_i$ is the measured BOLD signal for region $i$, $\epsilon_i$ is the white noise, $\sigma_i^2$ is the noise variance, and $I_{N \times N}$ is an identity matrix. Substituting Eqn. (7) into Eqn. (6) leads to the following expression for the measured fMRI signal:

$$(e^{2\pi i \frac{m}{N}} - 1)\frac{\hat{y}}{T} = A\hat{y} + C\hat{u} + \nu, \qquad (8)$$

where $\nu$ is a noise vector. By introducing a partial independence assumption that treats $\nu_i$ as an independent random vector with a noise precision parameter $\tau_i$, we can reformulate Eqn. (8) as a standard multiple linear regression problem:

$$Y = X\theta + \nu, \quad \nu \sim \mathcal{N}(\nu; 0, \tau^{-1} I_{N \times N}), \qquad (9)$$

where $Y$ is the dependent variable, $X$ is the design matrix (set of regressors), and $\theta$ is the parameter vector. The above derivation has transformed the linear DCM in the time domain to a general linear model (GLM) in the frequency domain, leading to the following likelihood function[43]:

$$p(Y|\theta, \tau, X) = \prod_{i=1}^{R} \mathcal{N}(Y_i; X\theta_i, \tau_i^{-1} I_{N \times N}) \qquad (10)$$

with

$$
\begin{aligned}
Y_i &:= (e^{2\pi i \frac{m}{N}} - 1)\frac{\hat{y}_i}{T}, \\
X &:= [\hat{y}_1, \hat{y}_2, \dots, \hat{y}_R, \hat{h u}_1, \hat{h u}_2, \dots, \hat{h u}_K], \\
\theta_i &:= [a_{i1}, a_{i2}, \dots, a_{iR}, c_{i1}, c_{i2}, \dots, c_{iK}],
\end{aligned}
\qquad (11)
$$

where $Y_i$ is the Fourier transformation of the temporal derivative of the measured BOLD signal in region $i$, and $u_k$ represents the $k$-th experimental input. After selecting appropriate priors on the parameters and hyperparameters, one can obtain a generative model for the estimation of EC and inputs, which can be performed very efficiently using a set of analytical variational Bayes (VB) update equations. By turning off the experimental inputs, rDCM can be applied to the rs-fMRI data. For details, see Frassle et al.'s articles[43,45,46].

## Metrics of effective connectome

The effective connectome is evaluated at the whole-brain, network, and regional levels. The mean whole-brain EC is calculated as the average of all $400 \times 400$ connections, while the mean whole-brain excitatory (inhibitory) EC is computed as the average of all positive (negative) EC. The within-network EC of a specific network is calculated as the average of all EC within that particular network, while the between-network EC of a specific network is computed as the average of all EC between that particular network and all other networks. The excitatory and inhibitory within-network (or between-network) EC is computed using positive and negative EC, respectively. The mean whole-brain within-network EC is computed as the average of within-network EC over the seven functional networks, and the mean whole-brain between-network EC is calculated as the average of between-network EC over the seven functional systems. Based on Sun et al.[21], we define the network segregation index (NSI) as the difference between the within-network EC ($EC_W$) and the between-network EC ($EC_B$) normalized by the within-network EC for a particular network:

$$NSI = \frac{EC_W - EC_B}{EC_W}. \qquad (12)$$

Similarly, the mean NSI is defined as the difference between the mean within-network EC and the mean between-network EC normalized by the mean within-network EC.

We also define a network integration index (NII) as the absolute sum of excitatory between-network EC and inhibitory between-network EC, so that inhibitory between-network EC also contributes to NII. The mean NII is calculated as the average NII among the seven networks, and the NII is normalized by the maximal NII value. On the regional level, the nodal EC strength (ECS) is defined as the sum of all incoming EC to a particular region, and the excitatory (inhibitory) nodal ECS is defined as the sum of all incoming positive (negative) EC. The network ECS is calculated as the average of nodal ECS within a particular network, and the mean network ECS is the average ECS of all seven networks. We use K-means clustering[96] to classify the 400 nodal ECS time series into five distinct clusters. We vary the number of K from 2 to 10 and identify the optimal number that gives the most meaningful and interpretable distinct clusters; increasing the optimal number of clusters leads to clusters with similar characteristics.

## Growth modeling

We utilize generalized additive mixed models (GAMMs) to fit the developmental trajectories of various metrics of effective connectome (implemented in the "gamm4" package in R). GAMMs are an extension of generalized additive models (GAMs) by incorporating random effects, which have been used widely to model correlated and clustered responses[97] as well as the nonlinear brain developmental trajectories[98-100]. The non-parametric GAMMs are adopted in this study as they show superiority in modeling brain developmental trajectories compared with parametric models[98]. Non-parametric GAMMs are formulated as

$$y_i(t) = f_1(t) + f_2(t) * \text{sex}_i + \alpha_i + \beta_i + e_i(t), \qquad (13)$$

where $y_i(t)$ is the metric of effective connectome for subject $i$ at age $t$, $f_1(t)$, and $f_2(t)$ are cubic spline functions, $\text{sex}_i$ denotes the sex of subject $i$ (1 for males and $-1$ for females), $\alpha_i$ denotes the random intercept effect for subject $i$, $\beta_i$ denotes the random intercept for the site of subject $i$, and $e_i(t)$ represents the random Gaussian noise associated with subject $i$ at age $t$. Note that the incorporation of the random site effect in the model serves to eliminate the site effects in the data (BCP, HCP-D, HCP-YA, and HCP-A). To better characterize the age-dependent EC changes during infancy and early childhood, we transform the age into a log scale ($t = \log_2(\text{age} + 1)$). For the cubic spline functions, we vary the maximal degree of freedom from $k = 5$ to $k = 14$ and select the model that gives the lowest Bayesian information criterion (BIC). To obtain sex-independent trajectories, we control the sex confounder by setting $\text{sex}_i$ to 0.

## Estimating peak and growth rate

The peak of a fitting curve is identified by locating the local maximum during the lifespan, and the corresponding age is identified as the peak age. The local maximum should be larger than its two neighboring samples, and the endpoints are excluded. If there are multiple local maxima, we take the largest maximum as the peak. For the inhibitory (negative) EC curves, we first flip the sign of the metric and use the same procedure to identify the negative peak. The 95% confidence intervals (CI) of the estimated peak age are calculated through a longitudinal bootstrapping method[101]. 20,000 samples from the age smooth term coefficient posterior distribution are drawn, reflecting model uncertainty in the estimates. For each of these possible realizations of the age-metric relationship, the longitudinal fit is recomputed, and the age at which each fit reaches its maximum is identified. The 95% CI is then computed based on this distribution of maximum ages. The growth rate is obtained by computing the first derivative of the growth curve using the Euler approximation. The 95% CI of the fitted curves is computed as

$$CI = y_{\text{pred}} \pm \text{qnorm}(1 - \alpha/2) * y_{\text{se}}, \qquad (14)$$

where $y_{pred}$ are the predicted values of the metric of effective connectome and $y_{se}$ are the standard error values determined using the 'predict()' function in R. 'qnorm$(1 - \alpha/2)$' is the critical value with $\alpha$ set to 0.05.

## Statistics and reproducibility

Since the peak ages of nodal ECS do not conform to normal distribution, we use the two-sided Wilcoxon rank sum test, a non-parametric version of the two-sample $t$-test, to compare the peak ages of nodal ECS between different functional networks (the sample size N is the number of regions within each network which is different for the two samples (ranging from 26 to 91)). In addition, we use the two-sided Wilcoxon signed-rank test, a non-parametric statistical test used to compare two related samples, to compare the standard deviation (SD) of nodal ECS before 12 years old and after 20 years old for each functional network (the sample size $N$ is the number of regions within each network which is the same for the paired samples). Multiple comparisons are corrected by controlling the false discovery rate (FDR)[102] at a significance level of $p < 0.05$. We choose the FDR correction as it balances statistical power and type I error[102]. We also report the Cohen's d effect size, which is determined by calculating the mean difference between the two compared groups and then dividing the result by the pooled standard deviation. Our results are highly reproducible following the methods and analysis approaches described above.

## Reporting summary

Further information on research design is available in the Nature Portfolio Reporting Summary linked to this article.

## Data availability

The source data for Figs. 1–5 are available in Supplementary Data 1–5. The Lifespan HCP fMRI data is publicly available via the National Institute of Mental Health data archive (NDA, http://nda.nih.gov). All the data are deposited under the Connectome Coordination Facility repository with the following Collection IDs (BCP: #2848; HCP-D: #2846; HCP-YA: #2825; HCP-A: #2847). All other data are available from the corresponding author on reasonable request.

## Code availability

A MATLAB implementation of the rDCM model is available as open source code in the Translational Algorithms for Psychiatry-Advancing Science (TAPAS) toolbox (www.translationalneuromodeling.org/tapas).

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

## Acknowledgements
This work was supported in part by the United States National Institutes of Health (NIH) through grants R01 MH125479, R01 EB008374, R01 EB035160, R01 MH133836, and R21 AG083589.

## Author contributions
G.L.: methodology, investigation, visualization, data curation, writing—original draft, writing—review and editing. K.M.H.: data curation, methodology, software, writing—review and editing. K.-H.T.: data curation, methodology, software, writing—review and editing. H.P.T.: data curation, methodology, software, writing—review and editing. G.L.: software, resources. W.L.: resources. S.A.: software, data curation, resources, writing—review and editing. P.-T.Y.: conceptualization, supervision, funding acquisition, investigation, validation, writing—review and editing.

## Competing interests
S.A. is an Editorial Board Member for *Communications Biology*, but was not involved in the editorial review of, nor the decision to publish this article. All other authors declare that they have no competing interests.
