## [Transparent Peer Review file · Communications Biology]

The Effective Connectome Over a Century of Human Life

Corresponding Author: Professor Pew-Thian Yap

Version 0:

Reviewer comments:

Reviewer #1

(Remarks to the Author)

In this study, the authors used a large sample of resting-state fMRI data from the BCP and Lifespan HCP datasets to map effective connectivity development across the human lifespan at the global, network, and regional levels. Their findings are intriguing. For example, regional effective connectivity strength is more variable in early life compared to later life stages, and its development follows a spatial constraint along the sensorimotor-association gradient, driven primarily by inhibitory interactions. While the lifespan development of functional organization is an important and timely topic, this manuscript seems somewhat rushed, and several key issues require further attention.

Major issues:

1. The benefits of effective connectivity analysis are unclear. The Introduction only briefly states that effective connectivity could provide mechanistic insights without giving details. Please clarify which specific aspects (e.g., excitatory and inhibitory interactions) can be further inferred from effective connectivity analysis. In addition, please compare the current findings with previous lifespan studies of functional connectivity in the Discussion, which would help highlight both the consistency and novelty of these findings.
2. Figure 1 shows relatively low mean values for whole-brain EC, within-network EC, and between-network EC. This raises concerns about the significance of age-related changes in these values. Could the authors address these observations in the Discussion? In addition, some developmental trajectories exhibit complex shapes, such as the orange curves in Figure 2a. Please verify the reliability and robustness of the trajectories reported here.
3. The authors applied regression dynamic causal modeling (DCM) to estimate effective connectivity but did not explain why this approach was superior to other models. On page 12, the transition from Eq. (3) to Eq. (4) is unclear; more explanation is needed regarding changes in the frequency term.
4. Most results appear to be derived qualitatively from visual inspection, although this is not explicitly stated. For example, it is unclear how the three major patterns of developmental trajectories for nodal effective connectivity were identified (page 6). Meanwhile, it is also unclear how to infer the differences in peak age between the association and primary systems (page 8).
5. The Desikan-Killiany atlas was used for the node definition, but its anatomical basis may reduce the functional homogeneity of the time series within each node. Please explain the choice of this atlas and confirm whether the main findings hold for functionally defined parcellations.
6. All ROIs were grouped into six functional networks instead of the original seven systems in Yeo et al. (2011). Specifically, the dorsal attention network was integrated into the frontoparietal network, while the ventral attention network, renamed the salience network, was classified as part of primary sensory systems. The rationale for these decisions is unclear.

Minor issues:

7. Some statements in the Discussion lack supporting results. For example, references to “(< 12) years” and “before 12 years of age” are repeated in the first paragraph. Why not mention 11 or 10? Please clarify how these conclusions were reached.
8. The nodal effective connectivity of the visual region peaks at a later age (e.g., 43.6 years), which is unusual. Could the authors clarify the biological interpretation of this finding?
9. In the data preprocessing, only high-pass filtering was applied, with no low-pass filtering. In addition, all fMRI data were analyzed in native space without explanation. Please provide a rationale for these preprocessing strategies.
10. Regarding the head motion control. Although BCP data with high head motion were excluded, no similar controls were applied to the Lifespan HCP data. Several previous studies have addressed head motion concerns for the HCP dataset. To address potential confounding effects, please assess and report head motion (e.g., FD) for scans in the Lifespan HCP dataset and verify whether the extent of head motion is age-dependent.
11. For Figures 1 and 2, please explain the color-shaded areas accompanying the age-stratified developmental curves at the network level.

12. The fMRI data used here were obtained from the BCP and Lifespan HCP datasets. However, the BCP dataset is referenced inconsistently throughout the manuscript. In addition, two abbreviations "PFC" and "FPC" are used in the results to refer to the frontoparietal network. Some figure text and legends also have font sizes that are difficult to read.

Reviewer #2

(Remarks to the Author)

I have read and reviewed this work several times because the topic is, in my opinion, of great importance. The question of understanding whether the structure of the EC and FC from the fMRI signal at rest can be an indicator of cognitive complexity or normality is of utmost importance for updating the concept of cognitive assessment, which until now has been based on outcomes. I really liked the approach, and the proposals made in the work, such as some of the indicators (Network Integration Index) for the study of the longitudinal system of networks. As I believe that the work deserves to be published, I will limit myself to raising some questions that could contribute to improving it.

1. It is not clear how the fact that the data comes from different databases is resolved. Adding results from different machines is often a source of variability.
2. No information is given about the cognitive state of the people analyzed. Normality is mentioned, but in older adults this is much more difficult to assume. Without this accreditation it seems easy to question some results.
3. There is no longitudinal analysis in the strict sense, although this would not be strictly necessary. However, it should be made clear how the analysis is managed. There is a gap between the analysis procedures and the results. For example, there is not enough data on the results of the DCM regression.
4. Since the outline of the work is not the usual one, it makes it somewhat difficult to read the different sections in an orderly manner.
5. There is no cognitive assessment that guarantees false negatives.

Version 1:

Reviewer comments:

Reviewer #1

(Remarks to the Author)

The authors have addressed most of my comments, and the manuscript has been greatly improved.

Nevertheless, one issue warrants further clarification: the discrepant peak age of visual regions observed between the network and nodal levels. While the authors have addressed this discrepancy from multiple perspectives, their explanations appear fragmented and lack coherence. Thus, it remains challenging to integrate different perspectives and derive a definitive, biologically interpretable conclusion regarding whether the maturation of effective connectivity in visual regions occurs earliest or latest.

Reviewer #2

(Remarks to the Author)

After reading the second version, I consider that the authors have responded adequately to my observations.

Responses to Reviewers' Comments

We thank the two referees for their thorough, detailed and insightful reviews of our manuscript. The manuscript has been substantially revised to address the concerns and comments of the reviewers (the major changes are highlighted in **blue** in the manuscript text). The major changes include the following:

- We used one unified preprocessing pipeline and the same quality control procedure to all cohort data
- We used the Schaefer-400 (7 networks) functional atlas to extract the BOLD time series
- We defined a new network integration index (NII) in addition to the network segregation index (NSI)
- We used K-means clustering to classify the 400 time courses of nodal effective connectivity strength (ECS) into five distinct clusters and examined the cluster characteristics/distribution in detail
- We additionally studied the lifespan trajectories of inhibitory EC at both network and regional levels

With these major changes, we are delighted to report that the major findings of this study remain qualitatively unchanged, which demonstrates that our results are highly robust and reliable. In the meantime, the use of a much finer functional atlas and the additional analysis reveal deeper mechanistic insights of EC development over the lifespan.

We want to mention that with the major changes and additional analysis, the old figures have been reorganized/updated and two new figures are added. We included the new updated figures at the end of this response document and specify the changes we made.

We would like to offer our sincere thanks to the reviewers for their constructive criticisms, as we feel that the manuscript has been substantially improved by addressing their comments. Below, we use black text for the reviewers' comments and blue text for our responses.

Reviewer #1

Comment 1.0: In this study, the authors used a large sample of resting-state fMRI data from the BCP and Lifespan HCP datasets to map effective connectivity development across the human lifespan at the global, network, and regional levels. Their findings are intriguing. For example, regional effective connectivity strength is more variable in early life compared to later life stages, and its development follows a spatial constraint along the sensorimotor-association gradient, driven primarily by inhibitory interactions. While the lifespan development of functional organization is an important and timely topic, this manuscript seems somewhat rushed, and several key issues require further attention.

Thank you for your positive feedback and constructive comments. As mentioned above, we have made major changes and conducted additional analysis to improve our study. We have also substantially revised the manuscript to report new results and findings. We hope you find the manuscript improved.

Comment 1.1: The benefits of effective connectivity analysis are unclear. The Introduction only briefly states that effective connectivity could provide mechanistic insights without giving details. Please clarify which specific aspects (e.g., excitatory and inhibitory interactions) can be further inferred from effective connectivity analysis. In addition, please compare the current findings with previous lifespan studies of functional connectivity in the Discussion, which would help highlight both the consistency and novelty of these findings.

This is an excellent point. As suggested, we have clarified the specific aspects of mechanistic insights (i.e., distinct roles of excitatory and inhibitory neural interactions in development and aging) that can be further inferred from EC analysis in Introduction (Para. 3, Page 2). In addition, we have compared our findings with previous lifespan studies of functional connectivity (FC) in the Discussion (Para. 2, Page 16). Specifically, previous studies have reported heterogeneous FC development patterns, including linear increase or decrease, nonlinear U-shape or inverted U-shape. By comparison, we found consistent nonlinear inverted U-shape developmental pattern at the whole-brain, network, and regional levels (except a linear decrease pattern for regional ECS), consistent with a recent large-scale FC lifespan study (Sun et al., 2025), though we observed a much earlier maturation peak for the average EC connectome.

Comment 1.2: Figure 1 shows relatively low mean values for whole-brain EC, within-network EC, and between-network EC. This raises concerns about the significance of age-related changes in these values. Could the authors address these observations in the Discussion? In addition, some developmental trajectories exhibit complex shapes, such as the orange curves in Figure 2a. Please verify the reliability and robustness of the trajectories reported here.

The reason why we observed relatively low mean values for whole-brain, within-network and between-network EC is because EC follows a log-normal weight distribution and the predominance of weak connections reflects the low-frequency influence inferred from the resting-state fMRI (rs-fMRI) data which is dominated by low frequency components (Frassle et al., 2021). We also observed that the mean EC value is scaled by the network size (i.e., larger network leads to lower EC). We have explained this phenomenon in the Discussion (Para. 2, Page 16 and Para. 1 Page 17).

Regarding the complex shape (i.e., double peaks before 20 years) shown in the previous version, we did not observe this pattern in this new version. We suspect that the complex shape could be generated by the noise and the coarse anatomical Desikan-Killiany (DK) atlas (which could reduce the functional homogeneity of the time series within each node as the reviewer pointed out in Comment 1.5). Nevertheless, all the other previous patterns are replicated in the current analysis.

Comment 1.3: The authors applied regression dynamic causal modeling (DCM) to estimate effective connectivity but did not explain why this approach was superior to other models. On page 12, the transition from Eq. (3) to Eq. (4) is unclear; more explanation is needed regarding changes in the frequency term.

This is a good point. Based on the reviewer's comment, we have explained the rationale of using regression DCM (rDCM) to estimate EC in the Methods section (under the subsection "Regression DCM", Page 22). Specifically, rDCM has several main advantages including high computational efficiency, no reliance on structural information and EC estimation at the single-subject level. These advantages make rDCM an ideal approach for big dataset application. In addition, we have added more details on the mathematical derivation of rDCM (Pages 23).

Comment 1.4: Most results appear to be derived qualitatively from visual inspection, although this is not explicitly stated. For example, it is unclear how the three major patterns of developmental trajectories for nodal effective connectivity were identified (page 6). Meanwhile, it is also unclear how to infer the differences in peak age between the association and primary systems (page 8).

This is a very good point. Based on the reviewer's comment, we used a K-means clustering algorithm to group the 400 nodal ECS time courses into five distinct clusters, which greatly increases the rigor and interpretability of the results. Also, we have provided detailed explanations for the differences in peak age between the association and primary systems (Para. 2 & 3, Page 10; Para. 1 - 3, Page 11). Basically, this is because the primary systems contain mostly increase-dominating patterns with large peak ages while the association networks consist mostly of decrease-dominating patterns with small peak ages, forming a continuum of developmental curves along the sensorimotor-association (S-A) axis (i.e., large positive slopes change to large negative slopes). Notably, the peak age distribution becomes opposite for the inhibitory nodal ECS.

Comment 1.5: The Desikan-Killiany atlas was used for the node definition, but its anatomical basis may reduce the functional homogeneity of the time series within each node. Please explain the choice of this atlas and confirm whether the main findings hold for functionally defined parcellations.

This is an excellent point. Based on the reviewer's comment, we have replaced the Desikan-Killiany (DK) atlas with the Schaefer-400 functional atlas (Schaefer et al., 2018). Though all the major findings remain unchanged for the new results, the use of the Schaefer atlas has enabled us to generate more meaningful results and provide more mechanistic insights into EC development. For example, we are able to reveal a U-shape development pattern for the inhibitory EC and find that maturation of excitatory and inhibitory EC follows opposite hierarchical sequences. Please see text for details.

Comment 1.6: All ROIs were grouped into six functional networks instead of the original seven systems in Yeo et al. (2011). Specifically, the dorsal attention network was integrated into the frontoparietal network, while the ventral attention network, renamed the salience network, was classified as part of primary sensory systems. The rationale for these decisions is unclear.

This is a good point. Previously with the coarse DK atlas, we only identified two ROIs belonging to the dorsal attention network, so we merged them into the frontoparietal network as some studies suggested that the frontoparietal control network and the dorsal attention network can be grouped into a larger “executive control” network due to their similarities (Littow et al., 2015; Gratton et al., 2018). With the new Schaefer atlas, all ROIs were grouped into the seven canonical networks. Based on the reviewer’s comment, we now used “ventral attention network” instead of the “salience network”, though they are used interchangeably in the literature. Also, the dorsal attention and ventral attention networks can be interpreted as the sensorimotor processing systems as they are situated between the early sensory systems and upstream limbic, control and default networks, consistent with their role in bottom-up processing of sensory information (see the Discussion section of Wang et al., 2019). In the text, we now used “attentional networks” to refer to the dorsal attention and ventral attention networks.

Comment 1.7: Some statements in the Discussion lack supporting results. For example, references to “(< 12) years” and “before 12 years of age” are repeated in the first paragraph. Why not mention 11 or 10? Please clarify how these conclusions were reached.

We chose 12 years since it is the boundary age between childhood and adolescence. We also found that the network excitation ranking switches between the sensorimotor systems and high-order networks at around 12 years old (Fig. 4d). We want to highlight that nodal EC during early life (infancy and childhood) is more variable than later life during adulthood. To substantiate this claim, we computed the average standard deviation (SD) of the nodal ECS before 12 years and after 20 years, and found that the SD before 12 years old is significantly higher than that after 20 years old (Fig. 4b). Please see the text under the section “Evolution of nodal and network effective connectivity strength across the life cycle” (Pages 11 – 13).

Comment 1.8: The nodal effective connectivity of the visual region peaks at a later age (e.g., 43.6 years), which is unusual. Could the authors clarify the biological interpretation of this finding?

We agree that this is an intriguing finding, but can be explained well in this new version. First, at the network level, we observed that the visual network exhibits the earliest or the second earliest peak for both excitatory and inhibitory within-network/between-network EC (Fig. 1d-g). That means when the EC is separated into excitatory and inhibitory components, and averaged within or between networks, the visual network shows the earliest maturation. Second, at the regional level, when all the EC inputs (including within-network and between-network EC) to a region is summed up, the regions within the visual network show the largest maturation peaks among the 7 networks. In fact, all other sensorimotor processing networks (somatomotor, dorsal and ventral attentions) display larger peak ages than the association systems (Fig. 3e). This is because, as mentioned above, the sensorimotor networks contain mostly the increase-dominating patterns with large peak ages, while the association systems consist mostly of the decrease-dominating patterns with small peak ages. Such network-dependent evolution patterns form a continuum of developmental curves along the S-A axis (i.e., large positive slopes change to large negative slopes), consistent with previous findings (Luo et al., 2024; Sun et al., 2025). Third, when the nodal EC strength is separated into excitatory and inhibitory components, the visual network shows the earliest inhibitory EC peaks (Fig. S6), which may be more important (than the excitatory EC) to create the early critical period for development (Larsen and Luna, 2018; Reh et al., 2020). Lastly, existing experimental observations support the late maturation of visual system. This includes late development of the visual spatial integration function in humans that could extend well into childhood and possibly adulthood (Kovacs et al., 1999) and the protracted development of glutamatergic proteins in human visual cortex up to 40 years old (Siu et al., 2017). We have clarified these important points in the revised manuscript (both in Results and Discussion).

Comment 1.9: In the data preprocessing, only high-pass filtering was applied, with no low-pass filtering. In addition, all fMRI data were analyzed in native space without explanation. Please provide a rationale for these preprocessing strategies.

We applied a high-pass filter ($f > 0.001$ Hz) to remove low-frequency drifts. A low-pass filter was not used, as recent studies have shown that meaningful neural signals can exist in higher frequency ranges (0.1–0.25 Hz) (Chen and Glover, 2015). Additionally, ICA-AROMA denoising uses high-frequency contents as one of the criteria to remove motion-related components, reducing the need for explicit low-pass filtering.

Voxel-wise analyses are often conducted in standard space (e.g., MNI) to establish voxel correspondence across subjects. On the other hand, our analysis is performed on the cortical surface using vertex-wise measures, where each subject's surface reconstruction ensures one-to-one vertex correspondence across individuals. Conducting the analysis in native space preserves anatomical precision, avoids misalignment from volumetric warping, and eliminates interpolation artifacts. Our surface-based approach enables accurate group-level comparisons without transforming data to a common voxel grid. We have provided the rationale for these preprocessing strategies in the “Data Processing” section under Methods.

Comment 1.10: Regarding the head motion control. Although BCP data with high head motion were excluded, no similar controls were applied to the Lifespan HCP data. Several previous studies have addressed head motion concerns for the HCP dataset. To address potential confounding effects, please assess and report head motion (e.g., FD) for scans in the Lifespan HCP dataset and verify whether the extent of head motion is age-dependent.

Based on the reviewer's comment, we applied the same head motion control method to the Lifespan HCP data as to the BCP data. Specifically, samples with mean Power's frame-wise displacement (FD) exceeding 0.5 mm were excluded. While the BCP cohort showed a higher exclusion rate due to motion, among the scans that passed quality control, there was no significant difference in FD between different cohorts (Supplemental Fig. S7). Of the QC-passed scans (FD < 0.5 mm), there was a moderate correlation ($R = -0.3$) between FD and age (FD is higher when age is smaller), but the correlation was NOT significant. We have now accessed and reported FD from the motion parameters obtained via head motion correction for all cohorts in the “Quality Control” section in Methods.

Comment 1.11: For Figures 1 and 2, please explain the color-shaded areas accompanying the age-stratified developmental curves at the network level.

The shaded area denotes the 95% confidence interval of the normative trajectories. We have added this explanation in the captions of all relevant figures.

Comment 1.12: The fMRI data used here were obtained from the BCP and Lifespan HCP datasets. However, the BCP dataset is referenced inconsistently throughout the manuscript. In addition, two abbreviations “PFC” and “FPC” are used in the results to refer to the frontoparietal network. Some figure text and legends also have font sizes that are difficult to read.

We now clarified that BCP is part of the Lifespan HCP studies and referenced it consistently. We have also corrected the typo “PFC”. Moreover, we have enlarged the figure text and legends.

Reviewer #2

Comment 2.0: I have read and reviewed this work several times because the topic is, in my opinion, of great importance. The question of understanding whether the structure of the EC and FC from the fMRI signal at rest can be an indicator of cognitive complexity or normality is of utmost importance for updating the concept of cognitive assessment, which until now has been based on outcomes. I really liked the approach, and the proposals made in the work, such as some of the indicators (Network Integration Index) for the study of the longitudinal system of networks. As I believe that the work deserves to be published, I will limit myself to raising some questions that could contribute to improving it.

Thank you for your positive feedback and constructive comments.

Comment 2.1: It is not clear how the fact that the data comes from different databases is resolved. Adding results from different machines is often a source of variability.

We harmonized the data from different cohorts/sites by modeling the random site effect in the GAMM model (see “Growth Modeling” in the Methods).

Comment 2.2: No information is given about the cognitive state of the people analyzed. Normality is mentioned, but in older adults this is much more difficult to assume. Without this accreditation it seems easy to question some results.

This is a good point. We have now explicitly stated that “All the subjects are healthy and cognitively normal who represent the typical developing and aging samples and are free to major neurological and psychiatric disorders. In particular, for subjects 60 years and older, cognitive assessments were conducted to exclude participants with impaired cognitive abilities” (see “Imaging Datasets” in Methods, Page 21).

Comment 2.3: There is no longitudinal analysis in the strict sense, although this would not be strictly necessary. However, it should be made clear how the analysis is managed. There is a gap between the analysis procedures and the results. For example, there is not enough data on the results of the DCM regression.

We included longitudinal data from BCP (see “Imaging Datasets” in Methods). The longitudinal data was incorporated by modeling Subject ID as a random intercept (see “Growth Modeling” in Methods). We now added the visualization of the mean EC matrix (from regression DCM) during different developmental stages in Supplementary Fig. S2.

Comment 2.4: Since the outline of the work is not the usual one, it makes it somewhat difficult to read the different sections in an orderly manner.

We have substantially revised the text of the manuscript and hope it can be read smoothly.

Comment 2.5: There is no cognitive assessment that guarantees false negatives.

For subjects older than 60, cognitive assessments were conducted to ensure the participating subjects were cognitively normal. See response to comment 2.2.

REFERENCES

- Chen JE, Glover GH (2015) BOLD fractional contribution to resting-state functional connectivity above 0.1 Hz. *Neuroimage* 107: 207-218.
- Frässle S, Harrison SJ, Heinzle J, Clementz BA, Tamminga CA, Sweeney JA, Gershon ES, Keshavan MS, Pearlson GD, Powers A, Stephan KE (2021) Regression dynamic causal modeling for resting-state fMRI. *Hum Brain Mapp* 42: 2159-2180.
- Gratton C, Sun H, Petersen SE (2018) Control networks and hubs. *Psychophysiology* 55: 13032.
- Kovács I, Kozma P, Fehér A, Benedek G (1999) Late maturation of visual spatial integration in humans. *Proc Natl Acad Sci USA* 96: 12204-9.
- Larsen B, Luna B (2018) Adolescence as a neurobiological critical period for the development of higher-order cognition. *Neurosci Biobehav Rev* 94: 179-195.
- Littow H, Huossa V, Karjalainen S, Jääskeläinen E, Haapea M, Miettunen J, Tervonen O, Isohanni M, Nikkinen J, Veijola J, Murray G, Kiviniemi VJ (2015) Aberrant functional connectivity in the default mode and central executive networks in subjects with schizophrenia - A whole-brain resting-state ICA study. *Front Psychiatry* 6: 26.
- Luo AC, Sydnor VJ, Pines A, Larsen B, Alexander-Bloch AF, Cieslak M, Covitz S, Chen AA, Esper NB, Feczko E, Franco AR, Gur RE, Gur RC, Houghton A, Hu F, Keller AS, Kiar G, Mehta K, Salum GA, Tapera T, Xu T, Zhao C, Salo T, Fair DA, Shinohara RT, Milham MP, Satterthwaite TD (2024) Functional connectivity development along the sensorimotor-association axis enhances the cortical hierarchy. *Nat Commun* 15: 3511.
- Reh RK, Dias BG, Nelson CA 3rd, Kaufer D, Werker JF, Kolb B, Levine JD, Hensch TK (2020) Critical period regulation across multiple timescales. *Proc Natl Acad Sci USA* 117: 23242-23251.
- Schaefer A, Kong R, Gordon EM, Laumann TO, Zuo XN, Holmes AJ, Eickhoff SB, Yeo BTT (2018) Local-global parcellation of the human cerebral cortex from intrinsic functional connectivity MRI. *Cereb Cortex* 28: 3095-3114.
- Siu CR, Murphy KM (2018) The development of human visual cortex and clinical implications. *Eye Brain* 10: 25-36.
- Sun L, Zhao T, Liang X et al (2025) Human lifespan changes in the brain's functional connectome. *Nat Neurosci* 28: 891-901.
- Wang P, Kong R, Kong X, Liégeois R, Orban C, Deco G, van den Heuvel MP, Thomas Yeo BT (2019) Inversion of a large-scale circuit model reveals a cortical hierarchy in the dynamic resting human brain. *Sci Adv* 5: eaat7854.

Changes of the main figures

Note that most of the results are qualitatively the same as the previous results. But with the new processing pipeline and a much more refined atlas, the results could be quantitatively different for the same EC metrics.

Figure 1. Global and network EC.

- This figure corresponds to previous Fig. 1.
- This figure shows very similar developmental patterns for both global and network EC as before.
- The subfigures of network segregation are moved to Fig. 2.
- The subfigures of excitatory and inhibitory within-network/between-network EC are added (Fig. 1d-g)

Figure 2. Network segregation and integration.

- This is a new added figure.
- It includes the evolution of network segregation index moved from the previous Fig. 1.
- It added the evolution of network integration index.
- The evolution of network segregation is similar to previous result.

Figure 3. Developmental patterns of nodal ECS.

- This is a new added figure.
- We use K-means to classify 400 fitted nodal ECS time courses into five distinct clusters.
- We plot the traces of nodal ECS in each cluster (Fig. 3a), global distribution of clusters (Fig. 3b), and cluster distribution within each network (Fig. 3c).
- We plot the correlation matrix of 400 fitted nodal ECS time courses (Fig. 3d)
- We visualize the peak age distribution of nodal ECS for each network (Fig. 3e).

Figure 4. Evolution of nodal ECS.

- This figure corresponds to previous Fig. 2.
- We replace the representative nodal ECS profiles in each network with the average nodal ECS profiles of each cluster in each network (Fig. 4a).
- We replace the brain surface plot of nodal development peak year with the standard deviation (SD) of nodal ECS profiles before 12 years old and after 20 years old for each network (Fig. 4b).
- The current network ECS result shows a clear switching at around 12 years old between sensory/attentional networks (VIS, SM, DAN and VAN) and high-order networks (FPC and DMN) (Fig. 4d).

Figure 5. Gradient of effective connectome.

- This figure corresponds to previous Fig. 3.
- This figure shows the same EC metric (gradient) as previous figure.
- With 400 nodal ECS (instead of 68) and the addition of a network (dorsal attention), the results are quantitatively different, but are qualitatively the same.

Figure 1 | Evolution of global and network effective connectivity (EC) over the lifespan. Normative trajectories (top) and velocity curves (bottom) of, **a**, mean whole-brain EC, **b**, mean within-network EC, and **c**, mean between-network EC stratified by sex. **d**, Developmental trajectories of the mean and network-specific excitatory within-network EC. **e**, Developmental trajectories of the mean and network-specific inhibitory within-network EC. **f**, Developmental trajectories of the mean and network-specific excitatory between-network EC. **g**, Developmental trajectories of the mean and network-specific inhibitory between-network EC. For a - g, the horizontal axis is in log scale and the short vertical bars indicate the peak ages. For a - c, the shaded area denotes 95% confidence interval of the normative trajectories, For d - f, the mean trajectory is manually lifted for better visualization. VIS: visual network, SM: somatomotor network, DAN: dorsal attention network, VAN: ventral attention network, LIM: limbic network, FPC: frontoparietal control network, DMN: default mode network.

Figure 2 | Evolution of network segregation and integration over the lifespan. **a**, Normative trajectories (left) and velocity curves (right) of the mean network segregation index (NSI) stratified by sex. **b**, Normative trajectories (left) and velocity curves (right) of the mean network integration index (NII) stratified by sex. **c**, Developmental trajectories of the mean and network-specific segregation index. **d**, Developmental trajectories of the mean and network-specific integration index. **e**, Cortical visualization of NSI at four selected time points. **f**, Cortical visualization of NII at four selected time points. For a - d, the horizontal axis is in log scale and the short vertical bars indicate the peak ages. For a and b, the shaded area denotes 95% confidence interval of the normative trajectories. For c and d, The mean trajectory is shifted upward for better visualization. VIS: visual network, SM: somatomotor network, DAN: dorsal attention network, VAN: ventral attention network, LIM: limbic network, FPC: frontoparietal control network, DMN: default mode network.

Figure 3 | Heterogeneous developmental patterns of nodal effective connectivity strength (ECS). **a**, Classification of 400 fitted nodal ECS time courses into five different clusters representing distinct temporal patterns. The average nodal ECS profile for each cluster is in bold shape. **b**, Distribution of nodal ECS among the five clusters. **c**, Percentage of each cluster in each functional network. **d**, Correlation matrix of the 400 nodal ECS time series. **e**, Distribution of the peak ages of nodal ECS in each functional network. For the boxplots, the central mark indicates the median, and the left and right edges of the box indicate the 25th and 75th percentiles, respectively. The whiskers extend to the most extreme data points not considered outliers, and the outliers are plotted individually using the “o” marker symbol. For a, the horizontal axis is in log scale. VIS: visual network, SM: somatomotor network, DAN: dorsal attention network, VAN: ventral attention network, LIM: limbic network, FPC: frontoparietal control network, DMN: default mode network.

Figure 4 | Evolution of nodal and network effective connectivity strength (ECS) over the lifespan. a, Average nodal ECS for each cluster in each functional network. **b**, Distribution of the standard deviation (SD) of nodal ECS profiles within the first 12 years and after 20 years old for each functional network. **c**, Nodal ECS visualized in the cortex at four selected time points (years 1, 9, 20 and 60). **d**, Developmental trajectories of the mean and network-specific ECS. The mean trajectory is shifted upward for better visualization. **e**, Normative trajectories of network ECS for each individual network stratified by sex. The shaded area denotes 95% confidence interval of the normative trajectories and the short vertical bars indicate the peak ages. For a, d, and e, the horizontal axis is in log scale. For the boxplots in b, the central mark indicates the median, and the bottom and top edges of the box indicate the 25th and 75th percentiles, respectively. The whiskers extend to the most extreme data points not considered outliers, and the outliers are plotted individually using the “o” marker symbol. VIS: visual network, SM: somatomotor network, DAN: dorsal attention network, VAN: ventral attention network, LIM: limbic network, FPC: frontoparietal control network, DMN: default mode network.

Figure 5 | Gradient of effective connectome. **a**, Nodal ECS gradient. Top: First principal component (loading) from the principal component analysis (PCA) on nodal ECS trajectories; Middle: Network-averaged loading; Bottom: Cortical visualization of nodal ECS loading. **b**, Network-averaged nodal ECS trajectories. **c**, Excitatory nodal ECS gradient. Top: First principal component from the PCA on excitatory nodal ECS trajectories; Middle: Network-averaged loading; Bottom: Cortical visualization of excitatory nodal ECS loading. **d**, Network-averaged excitatory nodal ECS trajectories. **e**, Inhibitory nodal ECS gradient. Top: First principal component from the PCA on inhibitory nodal ECS trajectories; Middle: Network-averaged loading; Bottom: Cortical visualization of inhibitory nodal ECS loading. **f**, Network-averaged inhibitory nodal ECS trajectories. For **a**, **c** and **e** (middle), error bars denote standard error. For **b**, **d**, and **f**, the horizontal axis is plotted in log scale. VIS: visual network, SM: somatomotor network, DAN: dorsal attention network, VAN: ventral attention network, LIM: limbic network, FPC: frontoparietal control network, DMN: default mode network.

Response to reviewers' comments

We deeply appreciate the positive feedback from the reviewers on our revised manuscript. We have addressed the remaining concern from reviewer #1. Below, we use black text for the reviewers' comments and blue text for our responses.

Reviewer #1

The authors have addressed most of my comments, and the manuscript has been greatly improved.

Thank you for your recognition and positive feedback.

Nevertheless, one issue warrants further clarification: the discrepant peak age of visual regions observed between the network and nodal levels. While the authors have addressed this discrepancy from multiple perspectives, their explanations appear fragmented and lack coherence. Thus, it remains challenging to integrate different perspectives and derive a definitive, biologically interpretable conclusion regarding whether the maturation of effective connectivity in visual regions occurs earliest or latest.

This is an excellent point, and we acknowledge that our previous response/revision was not sufficient. To further address the reviewer's concerns, we have conducted additional analysis that explained the discrepancy. **First**, for the inhibitory effective connectivity (EC), the sensory/attentional networks (including the visual network) show consistent earliest maturational peaks at both the network (**Fig. 1e, g**) and nodal levels (**Supplemental Fig. S6e**). **Second**, for the excitatory EC, the visual network displays the earliest within-network EC peak (**Fig. 1d**) and the second earliest between-network EC peak (**Fig. 1f**), whereas at the nodal level, the visual network shows the latest maturational peak (**Supplemental Fig. S5e**). Such discrepancy arises because the nodal ECS is defined as the sum of all incoming EC to a particular region, while the between network EC includes both incoming and outgoing between-network EC (see the new **Supplemental Fig. S7a, b**, attached at the end of this document). To elucidate the difference, we charted both incoming excitatory between-network EC and outgoing excitatory between-network EC (**Supplemental Fig. S7c, d**). Indeed, while the visual network maintains its earliest peak for outgoing excitatory between-network EC (i.e., from visual network to other networks), it shows the latest peak for the incoming excitatory between-network EC (i.e., from other networks to the visual network), consistent with the largest peak age for the excitatory nodal ECS. **Lastly**, when the excitatory and inhibitory EC is combined (i.e., the overall nodal ECS), the differential maturational peak ages between sensory/attentional networks and association systems are more prominent (**Fig. 3e**) because the decreased inhibitory EC in the sensory/attentional networks will contribute to the continued increase of nodal ECS during

adolescence and adulthood (as they have earlier maturation peaks). **In summary, the visual network matures the earliest for inhibitory EC, within-network EC and outgoing between-network EC, while it matures the latest for incoming excitatory EC.**

We have added the following paragraph in the Result section (the last paragraph of the subsection “Heterogeneous developmental patterns of nodal effective connectivity strength”).

“The smaller peak ages of the inhibitory nodal ECS in the sensory/attentional networks compared to the association systems (Supplementary Figure S6e) were consistent with their earlier maturation peaks in both within-network and between-network inhibitory EC (Figures 1e, g). However, the largest peak age of the excitatory nodal ECS in the visual network (Supplementary Figure S5e) seemed to be contradictory to its earliest (or second earliest) peaks in within-network and between-network excitatory EC (Figure 1d, f). One major difference was that nodal ECS reflected the sum of only incoming EC to a particular region (Supplementary Figure S7a), while between-network EC contained both incoming (from other networks to a particular network) and outgoing (from a particular network to other networks) between-network EC (Supplementary Figure S7b). To examine such potential differences, we charted both incoming and outgoing excitatory between-network EC (Supplementary Figure S7c, d). Indeed, while the visual network maintained its earliest peak for the outgoing between-network EC, it exhibited the latest peak for the incoming excitatory between-network EC, consistent with its latest maturation for the excitatory nodal ECS. Thus, the developmental maturation of nodal and network EC depends not only on its excitatory or inhibitory nature, but also its direction, fundamentally different from FC maturation which has no direction and does not distinguish between excitation and inhibition.”

We also added the following text to the Discussion section (end of Para. 9):

“It should be noted that nodal ECS reflects the sum of incoming EC to a particular region. On the network level, the visual network exhibits the earliest (or second earliest) maturation peak for within-network and between-network excitatory EC (Figures 1d, f). Such discrepancy (between regional and network maturation) can be explained by the fact that incoming between-network EC (consisting of incoming nodal EC) in the visual network displays more prolonged maturation than outgoing between-network EC (Supplementary Figure S7c, d), as discussed above.”

Reviewer #2

After reading the second version, I consider that the authors have responded adequately to my observations.

Thank you for your recognition and positive feedback.

Figure S7 | Differential peak ages for incoming and outgoing between-network EC. **a**, Diagram illustrating nodal effective connectivity strength (ECS) which is the sum of all incoming EC to a particular region. **b**, Diagram illustrating within-network EC, incoming between-network EC and outgoing between-network EC for the visual network. **c**, Developmental trajectories of the mean and network-specific incoming excitatory between-network EC. **d**, Developmental trajectories of the mean and network-specific outgoing excitatory between-network EC. For **c** and **d**, the mean trajectory is manually lifted for better visualization, the horizontal axis is in log scale, and the short vertical bars indicate the peak ages. VIS: visual network, SM: somatomotor network, DAN: dorsal attention network, VAN: ventral attention network, LIM: limbic network, FPC: frontoparietal control network, DMN: default mode network.